# Pb clustering and PbI$_2$ nanofragmentation during methylammonium lead iodide perovskite degradation

Alessandra Alberti [1], Corrado Bongiorno[1], Emanuele Smecca[1], Ioannis Deretzis [1], Antonino La Magna[1] & Corrado Spinella[1]

Studying defect formation and evolution in MethylAmmonium lead Iodide (MAPbI$_3$) perovskite layers has a bottleneck in the softness of the matter and in its consequent sensitivity to external solicitations. Here we report that, in a polycrystalline MAPbI$_3$ layer, Pb-related defects aggregate into nanoclusters preferentially at the triple grain boundaries as unveiled by Transmission Electron Microscopy (TEM) analyses at low total electron dose. Pb-clusters are killer against MAPbI$_3$ integrity since they progressively feed up the hosting matrix. This progression is limited by the concomitant but slower transformation of the MAPbI$_3$ core to fragmented and interconnected nano-grains of 6H-PbI$_2$ that are structurally linked to the mother grain as in strain-relaxed heteroepitaxial coupling. The phenomenon occurs more frequently under TEM degradation whilst air degradation is more prone to leave uncorrelated [001]-oriented 2H-PbI$_2$ grains as statistically found by X-Ray Diffraction. This path is kinetically costlier but thermodynamically favoured and is easily activated by catalytic species.

[1] CNR-IMM, Zona Industriale Strada VIII n°5, 95121 Catania, Italy. Correspondence and requests for materials should be addressed to A.A. (email: alessandra.alberti@imm.cnr.it)

There is a general consensus in referring to organic–inorganic hybrid perovskites, with particular focus on MAPbI$_3$, as soft matter. The degrees of freedom of the organic cations inside the inorganic Pb–I cage and their bonding to the anions create a crystal-liquid duality[1,2] that makes the lattice dynamically rearranging at the local scale during short lapses of time. The structure is indeed measured as average response of the material. Implications of this dynamic disorder have been already envisaged on the carrier transport[3], on the self-healing of lattice defects[4] and on order-disorder lattice transition with temperature[1,5].

Similarly related to the dynamic nature of the ionic interaction, lattice instability rises substantial concerns on the material reliability for long-lasting operation[6].

In the MAPbI$_3$ lattice, the attitude of the methylammonium cations (MA$^+$) to dynamically interact with external agents (species, electric fields and light)[7] reduces the perovskite durability with augmented vulnerability at the tetragonal to cubic phase transition[8]. Although the lattice engineering of different kinds of perovskite has been producing significant improvements by mixing cations and anions of different nature[9–11], the pioneering MAPbI$_3$ is still deserving top records in Photovoltaics and is indeed worthy of further effort[12,13].

In reasoning about MAPbI$_3$ stabilising solutions, the focus has been going with progressively more strength towards the preservation of the grain surfaces via treatments[14,15], extra-molecules or additives[16–22]. Grain shells are sites for lattice instability and dynamic generation of defects. Nitrogen, largely used as inert species in the experimental setup[8,11,12,23–25], has been recently demonstrated to have a beneficial role in assisting a temperature-induced recovery of intrinsic MAPbI$_3$ lattice disorder at the grain shell and in stabilising under-coordinated cations (Pb$^{2+}$, MA$^+$) through van der Waals interactions[26].

A thorough knowledge of the local defect formation and evolution inside a MAPbI$_3$ matrix was for a long time limited by the material damage caused by the use of high-dose electron beams as probe. They can create artefacts or provide misleading information. Most of the literature was conscious of that and a long silence was covering new insights. The first light was shed by Rothmann et al.[27] that used a very low electron dose rate (~1 e$^-$/Å$^2$s) for transmission electron microscopy (TEM) analyses. In that paper, for the first time, the transition from tetragonal to cubic lattice arrangement in thin polycrystalline MAPbI$_3$ layers was investigated with interesting insights drawn on the existence of reversible twinned domains. Very recently, a second paper has been published based on TEM analyses[28], providing a detailed description of single-crystal MAPbI$_3$ degradation to PbI$_2$ via I$^-$- and MA$^+$-related defects formation and volatile species release.

The applicability of polycrystalline MAPbI$_3$ layers to technology indeed demands special care on the transformation at the grain boundaries (GBs) and at all the exposed surfaces in respect to the volume, since defects formation and their further evolution pilot the whole perovskite to a progressive and irreversible transition to PbI$_2$. The kind of defects, their location and evolution, their eventual passivation, their optical behaviour are topics of increasing interest.

Starting from the pioneering papers on TEM, in our paper we afford the issue of surface defect formation in polycrystalline MAPbI$_3$ films by in situ TEM investigation at low total electron dose unveiling breakthroughs on: a clustering phenomenon of Pb-based defects at the MAPbI$_3$ grain surfaces; their motion without bias and their progressive enlargement during time at the expenses of MAPbI$_3$ GBs; the antagonistic formation of PbI$_2$ and its structural coupling with the MAPbI$_3$ mother grain that produces a textured nano-fragmentation as in a relaxed hetero-epitaxy. The described phenomena are similarly observed in thin and thick samples (40–150 nm), with clustering more pronounced in samples with high surface to volume ratio.

As a bridge towards degradation under ambient conditions, we feature similar investigations on identical samples exposed to humid air to draw a general paradigm on the role of defects formation and evolution, and on the convenience of PbI$_2$ to arrange along 2H or 6H stacking during MAPbI$_3$ degradation.

Thirdly, a lighting up phenomenon of Pb-clustering that is blocked by PbI$_2$ formation is disclosed and paves the way to frame advanced stabilisation actions that can inhibit formation of defects at the GBs.

## Results

**Degradation by Pb-clustering**. A set of 40-nm thick perovskite layers was deposited at 70 °C by sequential sublimation of PbI$_2$ and methylammonium iodide (MAI) directly on a Cu carbon-coated grid. The thickness was chosen to enhance the response under low total electron dose condition. Immediately after deposition, the Cu carbon-coated grid was loaded into the vacuum chamber of the TEM and analysed. To set up the proper acquisition conditions, we moved from the protocol developed by Rothmann et al.[27] that defines proper electron dose rate and energy needed to preserve, at least for short-acquisition times, the initial structure of the material. To keep low the total energy dose per frame, we set up the acquisition time at 200 kV as low as 4 s in the 40-nm thick layer at dose rate of ~1 e$^-$/Å$^2$s. We further implemented the protocol by on purpose switching off the beam between two consecutive acquisitions in order to reduce material–beam interactions during time. Thereby, frames were acquired for 4 s each, along 1500 s of net exposure (excluding the time with the beam off). Our first intent is understanding the roles of e-beam and vacuum since the early stages of interaction.

Figure 1 shows a collection of images acquired during time. The first image represents the sample in fresh conditions. The layer is made of grains reciprocally matching along the GBs with diameter in the range 50–200 nm. Despite of the low deposition temperature used (70 °C), no pin-holes were found on the surface. The same was achieved on FTO/TiO$_2$ substrates. The surface of the grains is smooth and uniform. Some grains are dark as a result of highly scattering crystallographic planes, in Bragg conditions with respect to the e-beam. Those planes specifically identify the tetragonal phase of MAPbI$_3$ without any PbI$_2$ contribution, as will be focused by the further discussion of data. Average compositional inhomogeneities are excluded on the basis of diffraction and chemical analyses.

During subsequent acquisition, the boundaries between adjacent grains (GBs) are being progressively modified. A net change is detected after 12 s from the sample loading. The circled regions in Fig. 1b–d highlights, in fact, small and dark grains located at the (preferentially triple) GBs that grow during time. While growing, they move along the GBs leaving behind empty spaces (holes) that are white in the image. The final size of clusters is in the diameter range 10–15 nm. The process is sketched in Fig. 1f. The nano-grains at the GBs are Pb-clusters as testified by the EDX spectra acquired in scanning mode and shown in Fig. 1e (light blue profile). Although the beam size is comparable with the Pb-cluster size, contributions from the surrounding matrix cannot be fully avoided. This indeed produces a small iodine peak in the profile. The reference spectrum is instead taken far from the GBs inside the grain (red curve) and has indeed intense I-related peaks. Quantification of the Pb–I ratio into the grain is beyond the scope of this analysis and would require specific protocol. EDX was instead on purpose used to certify the initially unknown composition of the clusters formed at the interfaces. Pb-clusters are not observed in fresh

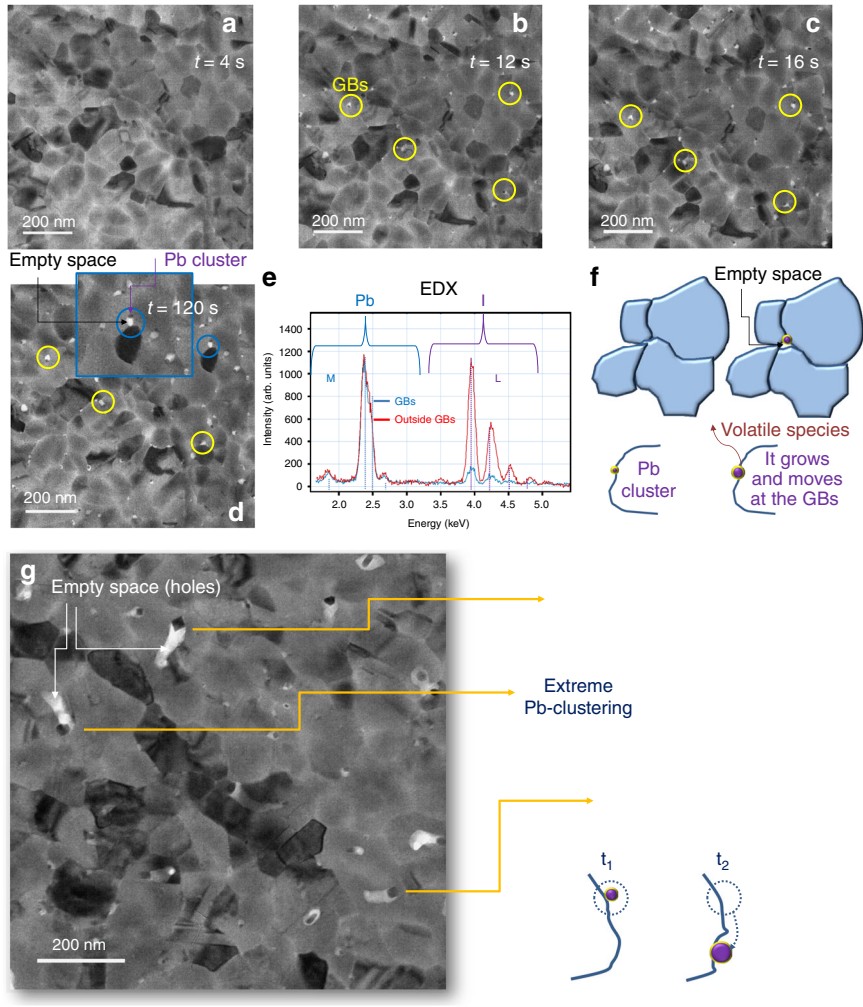

**Fig. 1** First step of MAPbI$_3$ transition: Pb clustering at the GBs. The layer thickness is 40 nm. **a–d** Plan-view TEM images sequence taken at a fixed sample position during time. For each image the electron dose rate was 1 e$^-$/A$^2$ s and the acquisition time was 4 s (e.g., 4 and 12 s indicates the first and the third frame acquired). Yellow circles indicate Pb-clusters at the GBs. The inset in **d** is a magnification to highlight a cluster moved along the BGs with a hole (empty space across the whole thickness) left behind (linked white area). **e** EDX profiles, normalised to the most intense Pb peak, collected at the end of this first step of transformation, inside and outside a Pb cluster (Shells: M and L of Pb and I, respectively). The vertical bars represent the expected relative intensities by Pb and I atoms. **f** Schematic showing the Pb nano-clusters growing at the perovskite grain boundaries and leaving empty spaces behind as in **d**. The original grains indeed start locally detaching. **g** Bright field TEM image depicting an extreme case of oversized Pb-clustering and related schematic. The arrows point out some big clusters feeding out the host matrix preferentially along the GBs

non-irradiated samples left under vacuum per days and then quickly irradiated for image acquisition for 4 s.

The increasing size of the Pb-clusters and the formation of holes at the GBs during time in a starting compact layer imply that MAPbI$_3$ is feeding the process and that, in concomitance, volatile species, containing iodine, are formed[29]. Although high reactivity between iodine and lead species at equilibrium in bulk conditions is expected, the observed iodine release and Pb clustering are, on the contrary, surface phenomena. In addition, vacuum conditions unbalance the local equilibrium towards the progressive volatilisation of species that leaves Pb atoms to easily aggregate on the surfaces. The preferential location of the Pb-clusters at the (triple) GBs reinforces the pivotal role of the surface and a reduced barrier for the overall process.

An extreme case is shown in Fig. 1g, taken by leaving, after analyses, the sample for two days under vacuum with the e-beam off. The image was taken in a fresh area of the sample. We believe that the oversize of some clusters in the image is initially triggered by electron exposure (even at low dose, low magnification) and

supplied by long-lasting pumping in vacuum. It is, in fact, expected that vacuum speeds up the kinetics of the process[11], and indeed our protocol included to switch the e-beam off among two acquisitions. Above the specific conditions that trigger Pb-defects formation (water, light, etc.), they are expected to aggregate with similar dynamics as under vacuum, thus preferring GBs for the high-lattice disorder[26,30], and with similar effects on the MAPbI$_3$ integrity. As a matter of fact, similar structural dynamics were previously observed for MAPbI$_3$ degradation under vacuum and air conditions using X-rays as probe for analyses[31,32]. Nonetheless, Pb-nanoclusters do not necessarily leave a measurable trace of them in conventional diagnostics (e.g., X-ray diffraction (XRD)) due to their small size and large dispersion.

**Degradation by nano-fragmentation**. It has been observed that the progression of the MAPbI$_3$ to Pb goes to saturation after ~120 s from loading, producing Pb-cluster with size of a few nanometres (10–15 nm). They do not notably evolve during

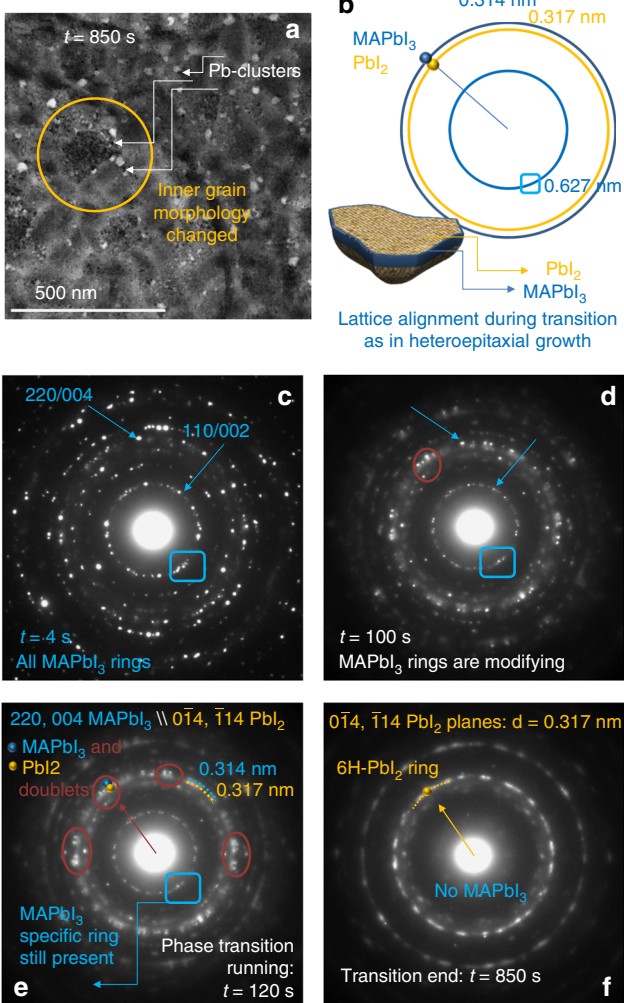

**Fig. 2** Second step of MAPbI₃ transition: from MAPbI₃ to fragmented 6H-PbI₂. The layer thickness is 40 nm. **a** Plan-view TEM image taken in the same region as in Fig. 1 after 850 s of electron exposure. At this time, the transition of MAPbI₃ to PbI₂ is completed. **b** Schematic on the nano-fragmentation of a single MAPbI₃ grain during its transformation: nano-PbI₂ grains are structurally connected to the mother perovskite grain as unveiled by the alignment of the diffracted spots. From **c-f** ED patterns (EDPs) tracing the evolution of the phase transition from MAPbI₃ to PbI₂ (volatile species are pumped away). The sequence is taken at fixed sample position over an area comparable to the image size in **a**. For each ED the acquisition time was 4s. Resulting main coupled interspacing distances of the two phases are summarised in **b** and labelled in light blue for MAPbI₃ and dark yellow for 6H-PbI₂. The specific ring at $d = 0.627$ nm univocally identifies the MAPbI₃ layer: it is used as marker to trace its full transformation to PbI₂. The spots at $d = 0.317$ nm are diagnostic of the 6H-PbI₂ polymorphism. MAPbI₃-PbI₂ doublets, some of them circled in dark orange in figures **d**, **e**, start forming after ~100 s (**d**): they are better resolved after 120 s (**e**). The blue boxes in **c-e** indicate the diagnostic ring of MAPbI₃: once fully lost, the transition is completed (**f**)

time. This transformation path is limited by a concomitant but slower second transition mechanism. This second stage is described in Fig. 2.

After ~100 seconds from loading, the grains' core starts radically changing its structure. The changes are gradual during time with final state reached after ~850 s. The plan-view TEM image shown in Fig. 2a, taken in the same region as the images in Fig. 1, is well representative of the final state of this second stage

of transformation. In the image, the grains, whose boundaries are not still changed from their starting configuration (see Fig. 1), are inside fragmented in small nano-grains (schematic in Fig. 2b) that creates roughening at the surface (further discussion in the next sections). On the contrary, the peripherical Pb-clusters, still located at the GBs, have size and position mostly unmodified as mapped in Fig. 1 at $t = 120$ s. Their further evolution was indeed blocked by the transformation of the whole grains.

To describe the changes occurring inside the grains during time, Fig. 2c–f show a sequence of electron diffraction patterns (EDPs) taken during time in the area of the sample shown in Fig. 2a. The starting arrangement of the lattice is provided in Fig. 2c wherein specific rings of the tetragonal phase (the two most intense labelled with the blue arrows) are easily associated to planes of the (110/001) family. The main planes of this family are represented in the schematic of Fig. 2b, together with the associated plane spacing: 0.627 and 0.314 nm for the (110)/(002) and the (220)/(004) planes, respectively. PbI₂ is fully absent in the starting pattern. During relatively short laps of time, we observe a progressive transfer of intensity from the original perovskite spots at position $d = 0.314$ nm towards a close contribution from PbI₂ at $d = 0.317$ nm. See Fig. 2d as representative of the phenomenon after 100 s of beam exposure (25th frame). The transition causes the (220/004) perovskite ring to get shaded contours (Fig. 2d). It is to notice that the consequent outbreak of doublets involves each perovskite contribution, as visible in Fig. 2e and as further elucidated through Fig. 3. It is additionally noteworthy that the transition leaves unchanged the direction of the $Q$ vector in the reciprocal space (i.e., the ED). The final PbI₂ has the 6H[33,34] lattice symmetry, univocally identified by the brilliant contributions at d = 0.317 nm, as shown in Fig. 2f. Correspondingly, the vanishing of the internal specific ring of the MAPbI₃ (110/002: $d = 0.314$ nm) traces a final full conversion within 850 s. A transition to 6H-PbI₂ (R-3m) was similarly observed by Chen et al.[28] on single-crystal MAPbI₃ samples. The authors describe this phase change as occurring through an intermediate transition characterised by I species leaving the perovskite lattice with consequent formation of superstructures (I- and subsequently MA-deficient). The marking features for this transient transformation to PbI₂ were not observed on our 40-nm thick layer. Instead, the coincidence of the final state in our polycrystalline layers and in single crystals[28] denotes a similar transition path of the MAPbI₃ core. Besides core transition, polycrystalline layers, that are better suited for applications, demand special care on the transformation at the GBs and surfaces that is the main focus of our investigation.

In this respect, we found that the transition of the grain core to 6H-PbI₂ has blocked further visible evolution of the Pb nano-clusters formed during the first stage from loading. We thus argue that the transition from MAPbI₃ to Pb is in competition with the one from MAPbI₃ to PbI₂. Both of them occur with release of mass. We additionally unveil that PbI₂ and Pb-clusters can share a boundary (plan-view image in Fig. 2a) without necessarily interacting or merging or further evolving into a unique phase.

We spent further effort in studying the origin of the preferential phase transition of the grain core to 6H-PbI₂. We start reconsidering in more details the MAPbI₃–PbI₂ doublet alignments, as those shown in the large area ED in Fig. 3a and taken after 200 s of beam exposure (some doublets are circled in white and light blue). We notice that main MAPbI₃ spots in the large area diffraction can be framed by squares (light blue) that are concentric with similar slightly inner squares of PbI₂ (dark yellow), with corners and half-edges occupied by doublets. Each inner square univocally identifies the −441/841/ −481 zone axis (crystallographic directions) of the 6H-PbI₂ since it is not expected in other polymorphisms. The identification of

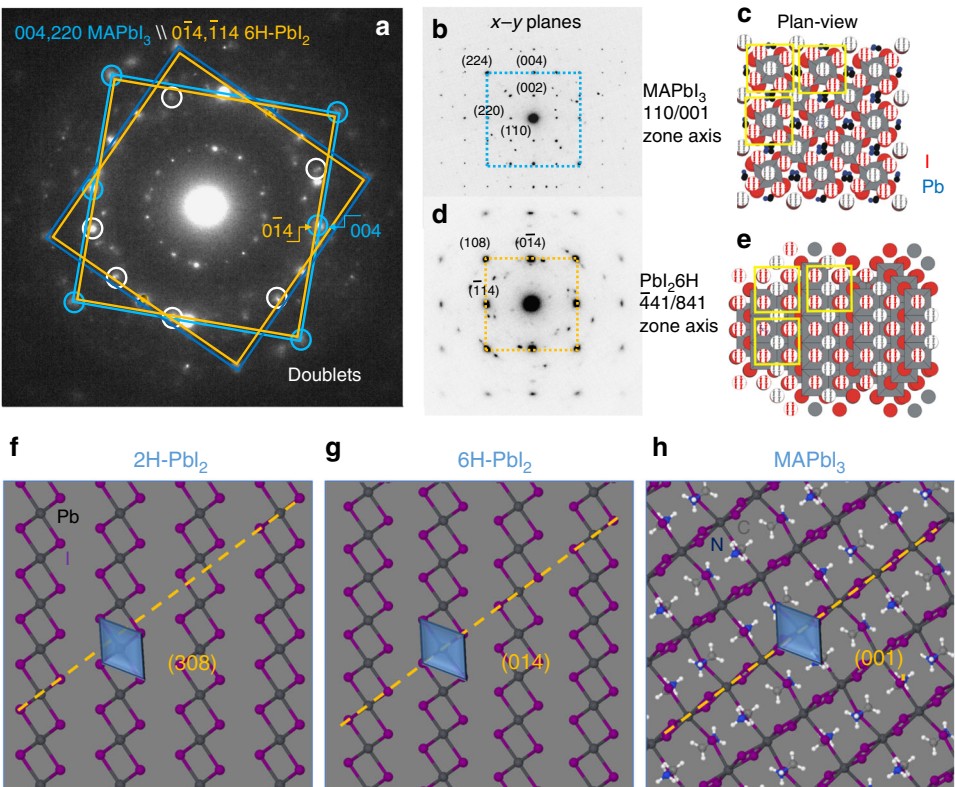

**Fig. 3** Three-dimensional structural coupling between MAPbI$_3$ and PbI$_2$. **a** Electron diffraction (integration time = 4s, $t$ = 200 s, total dose 200 e$^-$/A$^2$) taken in a smaller area than that used for Fig. 2a to reduce the number of grains and thus to highlight the lattice coupling at each doublet alignment (some doublets are circled in white or light blue). Light blue and dark yellow squares (also reproduced in **b** and **d**) connect spots of the same zone axis in the MAPbI$_3$ grain transforming into a corresponding 6H-PbI$_2$ grain: they focus on how one lattice changes into the other. **b**, **d** Selected area electron diffraction on MAPbI$_3$ and 6H-PbI$_2$ single grains and **c**, **e** simulated atoms alignment on the basal plane (dashed dots are coplanar atoms) showing MAPbI$_3$/PbI$_2$ mutual relationship. The layer thickness is 40 nm. Light yellow squares in **c**, **e** lie in the plane of the tetrahedron containing four iodine species and one lead atom: they highlight how the transformation preserves in-plane atomic configurations. **f**–**h** Schematic showing the atomic arrangement along diagnostic planes in MAPbI$_3$ and two observed polymorphisms (for MAPbI$_3$ and 6H-PbI$_2$ taken from the corner planes in **b** and **d**, respectively): moving from MAPbI$_3$ to 6H-PbI$_2$ is easier than to 2H-PbI$_2$ due to the closer atomic configuration. The yellow line is a guide for eyes to follow the octahedron alignment that is identical in the MAPbI$_3$ and 6H-PbI$_2$. Octahedra (represented in blue) are instead spatially shifted in nearby layers in the 2H-PbI$_2$ lattice with respect to the initial perovskite

this zone axis as belonging to 6H-PbI$_2$ is pivotal for data interpretation: misleading conclusions can arise by the incorrect attribution of the diffraction spots therein to MAPbI$_3$ that instead has planes at 0.314 nm and not at 0.317 nm. By discrete rotation of this coupled squares, the whole patterns can be represented. This implies that all the MAPbI$_3$ grains inside the analysed area of the sample change in similar 6H-PbI$_2$ grains. A difference is introduced by the initial rotational degree of freedom of the MAPbI$_3$ grains along the growth axis perpendicular to the sample surface.

The local EDPs taken on a MAPbI$_3$ grain before transformation (Fig. 3b) and on the corresponding PbI$_2$ grain after transition (Fig. 3d), locally reveal how (004/220) planes of the perovskite transform into (0–14/−114) planes of the by-product though the whole core. Those planes are aligned during transition as encountered in heteroepitaxial growths[35–38]. During transition, the single spots are smeared due to the nano-fragmentation of the matrix. This induces a broadening of the intensity distribution for size effects. Structure simulations in Fig. 3c, e show the atomic distribution and the relationship between atoms along the experimental zone axis and highlight a noteworthy similarity between MAPbI$_3$ and 6H-PbI$_2$. Moving from one phase to the other implies just a little change in the atomic configuration as

shown in Fig. 3g, h. Note, in particular, the octahedron alignment in the two structures with respect to what occurs in the 2H lattice (Fig. 3f). It is indeed intuitive that 6H formation is kinetically favoured.

Dark-field (DF) images were acquired during time selecting diagnostic diffraction spots. Figure 4 shows the staring (a) and the end (b) states in a sampled area that are representative of the whole sample behaviour. The circled area in the inset of the figure indicates the size of the objective aperture used for DF. The structural alignment of the two materials (doublet) allowed collecting information on the starting and end-states in the same sampled area by keeping fixed the slit position in the EDP. To read the image, bright areas in DF have the selected (inside the slit area) crystallographic planes in Bragg condition with respect to the beam. On this basis, we identify a starting large uniform grain of MAPbI$_3$ with random contours. After full transition to PbI$_2$, (~820 s), that is signed by the disappearance of the inner ring of perovskite at $d$ = 0.627 nm, the grains break into nano-aggregates of 6H-PbI$_2$. This results in a dotted black and white contrast at the nano-scale due to the fragmentation of the starting unique MAPbI$_3$ platform into slightly misaligned aggregates (slight zone axis tilt). This morphological transition is represented in the schematic of Fig. 4c. The preserved alignment of the PbI$_2$ planes

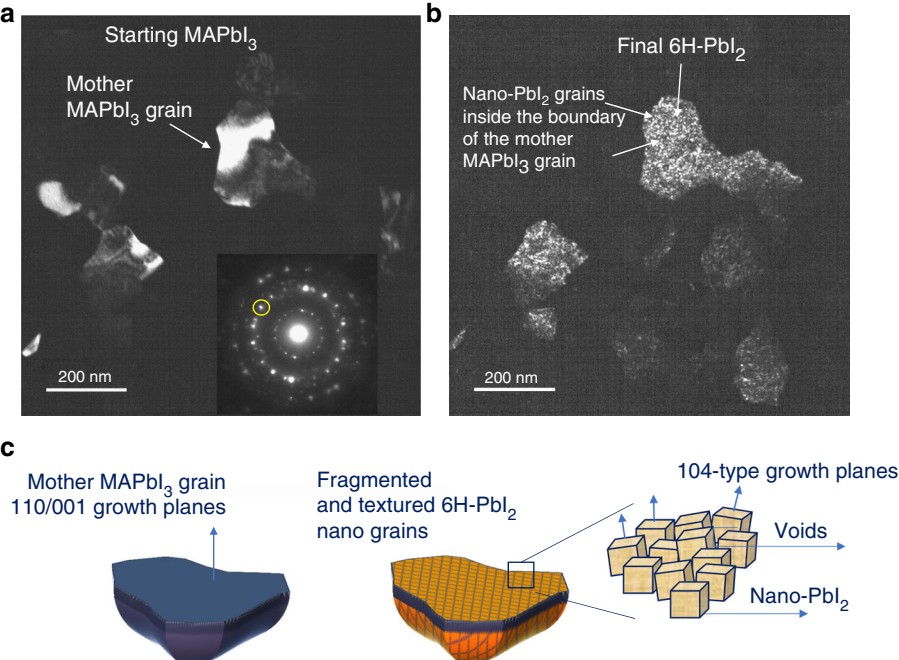

**Fig. 4** Fragmentation of the starting large MAPbI$_3$ grains into nano-grains of PbI$_2$. The starting layer thickness is 40 nm. **a** Fresh sample image taken in dark field (DF) by selecting the area of the EDP indicated in the inset by the yellow circle. **b** DF taken in the same region as in **a** with the aperture of the slit unchanged (see inset in a) after ~ 820 s (transition end). **c** Representation of the lattice alignment between MAPbI$_3$ and PbI$_2$ during transition at the grain core: it is highlighted that the growing planes (104 or the equivalent 1-14/014 are the same for all the nano-grains nucleated inside the boundary of a shared perovskite mother grain apart from a slight spread of their direction (represented by the vectors over the single grains in the right side schematic). The images are representative of the phenomenon all over the sampled area

to those of the starting mother MAPbI$_3$, occurring inside a window of tolerability within the zone axis spread, is often encountered after strain relaxation in heteroepitaxial growths[35].

**Degradation under air condition and its relation with TEM degradation.** A cross-link to the in-air degradation pathway was done through a focused experiment. To this purpose, a MAPbI$_3$ layer grown on a Cu-grid in the same deposition run as done for the sample TEM-degraded, was left in ambient air condition at 50–60% RH. After 5 days in air the sample colour fully tuned to yellow, providing a feedback of the occurred transition to PbI$_2$. Then, it was loaded into the TEM chamber and analysed under low-dose condition. Figure 5 compares the end-states reached through the two degradation paths. Both end states correspond to pure PbI$_2$. Figure 5a shows grains structurally fragmented, as discussed in Fig. 4, whose contours are invariant with respect to the starting perovskite grains. As a difference, the sample aged in air (Fig. 5b) has a layered architecture of [001] oriented sliced grains (c-axis oriented). This path is likely piloted by the presence of ambient water molecules that have a high capability to infiltrate the initial perovskite grains and to act as catalysts on the production of volatile species[11,25,26,31]. The in-air degradation is slower than in TEM degradation to allow water infiltration and release processes[11].

It has produced, among the allowed polymorphisms, the one indicated in the literature as the most stable at room temperature[39,40], namely the 2H (http://rruff.geo.arizona.edu/AMS/amcsd.php)[41]. The identification of the layered sequence of atoms in the sliced grains as following the P-3m1 (2H) arrangement is feasible on the basis of the selected area EDP in Fig. 5c. It was taken on a grain of the transformed material, as that shown in Fig. 5b, but is representative of most of the grain structure.

The EDP has a peculiarity that distinguishes this from the 6H polytype pattern. The peculiarity resides in the internal spots at plane distance of 0.395 nm. They are not expected in the 6H polymorphism (http://rruff.geo.arizona.edu/AMS/amcsd.php). The same planes could be found in 4H and 6R lattices, but with less intensity (2–3%$I_{max}$). Their dominance was statistically excluded on the basis of the large area XRD data that will be discussed in the next section. Moreover, 4H and 6R polytypes have twinned octahedron sequence around the c-axis, quite dissimilarly to what occurs in the MAPbI$_3$ lattice.

Although in the sample aged in air most of the grains are sliced along the (001) planes of the 2H structure that are parallel to the substrate, some other grains were less frequently found with the same zone axis as encountered during degradation at the TEM. To disentangle about this concomitance, large area XRD patterns were acquired and analysed to provide larger statistics on the lattice arrangement of the degradation products. To this purpose, different samples were prepared on glass substrates for large area X-ray analysis (1 cm × 1 cm) and measured after full degradation under ambient air plus eventual heating in the range 30–120 °C. Temperature was used to accelerate the degradation path[8,26]. The summarising scenario is shown in Fig. 6a–d. For a better figure readabiity, we clarify that the XRD patterns were taken in symmetric configuration and this implies that the peaks therein must be associated to planes lying parallel to the sample surface (Fig. 6b). We found that most of the samples exhibited a degradation path towards 2H-PbI$_2$, as shown in Fig. 6a by the intense peak at $2\theta = 12.67°$ and in agreement with the results in Fig. 5b, c. The result is independent of the temperature applied for degradation, indeed we do not discriminate the samples on the basis of their history but rather on the basis of their final state. We called this path A. We point out that, even though the peak at $2\theta = 12.67°$ can be exhibited by other polymorphisms, the cross correlation with the TEM pattern (Figs. 5c and 6e, inner hexagon)

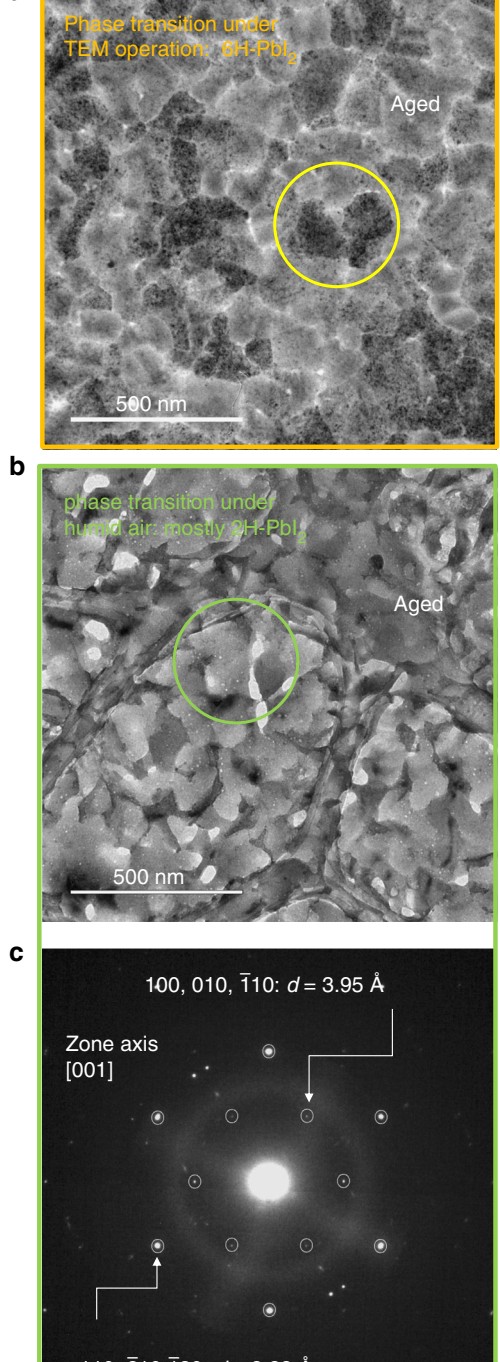

**Fig. 5** Degradation path to $PbI_2$ of $MAPbI_3$ deposited on grid and left under humid air. The layers are 40 nm thick. **a** Plan-view TEM image after full in-TEM degradation along the in situ time-resolved experiment of the on-grid-deposited $MAPbI_3$ layer (see Figs. 1 and 2). The TEM-degraded sample after ~850 s of beam exposure is fragmented into nano-aggregates of the 6H polytype as commented in Fig. 4. The yellow circle highlights 6H-$PbI_2$ grains with their peculiar morphological texture: note that the contour of the pristine $MAPbI_3$ mother grain is preserved. **b** Plan-view TEM image after full degradation under air condition for 5 days of a fresh on-grid-deposited $MAPbI_3$ layer. The sample is exfoliated in slices mostly having the (001) planes of the 2H polymorphism parallel to the surface. The green circle highlights 2H-$PbI_2$ grains with their peculiar sliced morphology. **c** Selected are ED taken mostly on the grain circled in **b**. It is characterised by a double hexagon with the labelled d-spacings identifying the 2H polymorphism. The diffuse ring is due to the carbon grid. This kind of ED is frequently encountered all over the sampled areas

mixture of polymorphisms (that could include the 4H) that share a common architecture in stacking rather than in separated grains. 6H inclusions in the frame of a 2H matrix is shown in Fig. 6d inside an in-air-degraded sample. 6H inclusions can be recognised by the peculiar texture, according to Figs. 2 and 4, as further confirmed by specific selected area diffraction as the one shown in Fig. 3d (squared diffraction). Figure 6e shows instead the selected area EDP of a 2H-$PbI_2$ grain with its peculiar double-hexagonal arrangement of spots (real and simulated). This pattern is easily distinguishable from a similar 6H-$PbI_2$ hexagonal configuration wherein instead the inner hexagon at $d = 3.95$ angstrom is missed (Fig. 6e, right panel).

We summarise that, under air condition and whatever the temperature used, the most convenient path for degradation of pure $MAPbI_3$ is towards the 2H polymorphism of $PbI_2$. In the large statistics, 6H-$PbI_2$ grains can be found with lower probability.

**Degradation in thick layers**. To more closely meet technological requirements, 150-nm thick layers of $MAPbI_3$ were grown on Cu C-coated grids simply by scaling up the deposit time. Scaling up the thickness is also used to frame the transition mechanisms in a more general paradigm. Due to the thickness, the acquisition time for TEM and ED was extended to 8 s/frame to improve the image quality. The total dose of electrons was low enough to not induce damage or artefacts at least in the first acquisition frames, as clearly stated in ref. [42]. Similarly to what found in the thin layer, the fresh thick sample is free from visible Pb clusters. It is made of large well-matching grains in the range of diameter 200–500 nm. In this sample, the surface/volume ratio is reduced compared to the 40-nm thick sample wherein the grain diameter is instead in the range 50–200 nm. Pb-clustering and hole formation at the GBs and surfaces is observed during the initial transformation step. Data are shown in Fig. 7b after an exposure time of 40 s (total dose = $40 e^-/A^2$). The phenomenon of Pb-clustering in thick $MAPbI_3$ layers deposited by solution processing was preliminarily reported in our previous publication[31]. Figure 7c shows, in addition, a phenomenon of neighbouring hole merging at the GBs during electron exposure as similarly observed in ref. [42] on $MAPbI_3$ layers deposited over Cu-grids by solution processing. In the statistics, hole merging can locally produce a detaching along a row of grains as shown in Fig. 7d, e. The similarity in the grain detaching over different deposition techniques sheds light on the phenomenon being uncorrelated with the specific growth method. Instead, differences in thickness, grain size, surface grain curvature or the presence of eventual contamination from the preparation procedure could introduce

allows a discrimination in favour of the 2H arrangement of the Pb–I octahedra (moreover, 2H is the most stable at RT). By Fig. 6a we further draw that inclusions of 6H-$PbI_2$ can be formed, as testified by the small peak detected at $2\theta = 28.23°$ ((104)-type planes, see figure 4c). This peak is not present in the 2H-$PbI_2$ pattern (see simulations in Fig. 6e). In some other $MAPbI_3$ samples, less statistically encountered, a predominance of the 6H polytype over the 2H can be found as shown in Fig. 6c. This countertrend is represented by the major relative intensity of the (104) peak of the 6H-$PbI_2$ over the (002) peak of the 2H polymorphism. This degradation path was called B. It is interesting to notice the unusual shape of this diagnostic peak centred at $2\theta = 28.23°$: it is broad at the basis as expected for a

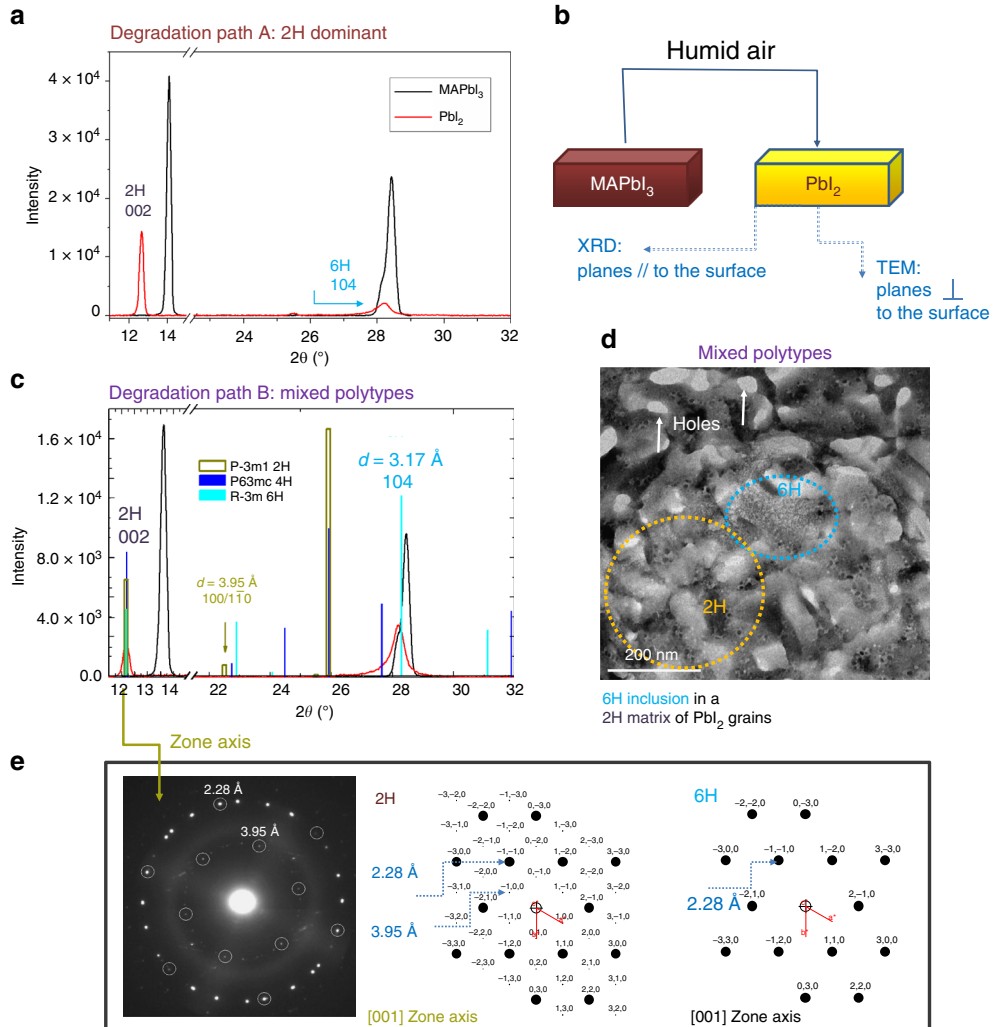

**Fig. 6** Large area statistics by XRD on PbI$_2$ after full in air degradation of MAPbI$_3$ layers deposited on glass. **a–c** $2\theta$-$\Omega$ symmetric X-ray diffraction patterns of fresh MAPbI$_3$ (black curves) layers fully converted into PbI$_2$ (red curves) under humid air conditions. In the statistics, the two degradation paths (A and B) represent possible scenarios for sample aged in air. Path A is the more frequently encountered. Path B is less frequent and represents a bridge toward the TEM-induced degradation. The vertical bars in **c**, that can be equally used for interpretation of the peaks in **a**, are markers for the different PbI$_2$ polymorphisms: light brown, blue and light blue lines stem for 2, 4, and 6H polytypes. Probed area = 0.2 cm$^2$. **b** Schematic of the experiment that combines TEM and XRD analyses to study parallel and perpendicular planes with respect to the sample surface. **d** BF-TEM analyses of an in air degraded sample on the Cu-grid, showing the inclusion of a 6H-PbI$_2$ grain (light blue circle) morphologically and structurally different from the rest of the matrix (integration time = 4 s). The dominance of the 2H arrangement is drawn on the basis of the specific spot at 3.95 angstrom in the EDPs reported in **e** wherein simulations discriminate between apparently similar 6 and 2H hexagonal patterns. In **d** holes are empty spaces left after volume contraction in the transition from MAPbI$_3$ to PbI$_2$ ($\Delta V/V$ ~50%); the yellow circle highlights typical 2H-PbI$_2$ sliced grains. 2H and 6H-PbI$_2$ grains are indeed morphologically distinguishable

differences from sample to sample. Figure 7e and its EDP in the inset additionally highlight a transformation of the MAPbI$_3$ grains towards PbI$_2$ through the formation of doublets similarly to what found in thin layers. A full transition is recorded, in that area of the sample, after ~1400 s of electron exposure (total dose ~1400 e$^-$/Å$^2$).

Similarly to what observed in the thin layers, Pb clustering and hole formation are the fastest processes that compete with the slower transformation of the grain core to PbI$_2$. Likely triggered at the surfaces, the transition proceeds with time to entirely transform the volume. The time to complete the transition, that was estimated in ~850 s for the 40-nm thick layer, is expanded by a factor ~2 in the 150-nm thick MAPbI$_3$ layer. Discrepancies of the degradation time from scaling with thickness could be related to the electron mean free path affecting the energy transferred to the material/unit volume. In addition, thickness related relaxation

of strain could impact on the transformation dynamics and therefore further effort is needed to disclose the detail of the core transition. Above this further refinement, the two-step degradation pathway is a general paradigm.

## Discussion

We draw the paradigm on a two-step transformation pathway of MAPbI$_3$ layers : Pb-clustering and phase transition to PbI$_2$. The paradigm is validated in the range of thickness from 40 to 150 nm but it could be further extended up to 300 nm.

The full description is provided by time-resolved experiments done at the TEM under mild (1 e$^-$/A$^2$ s and short-acquisition time) and intermitting e-beam conditions in order to limit the effect of the electron dose and exposure on the material structure.

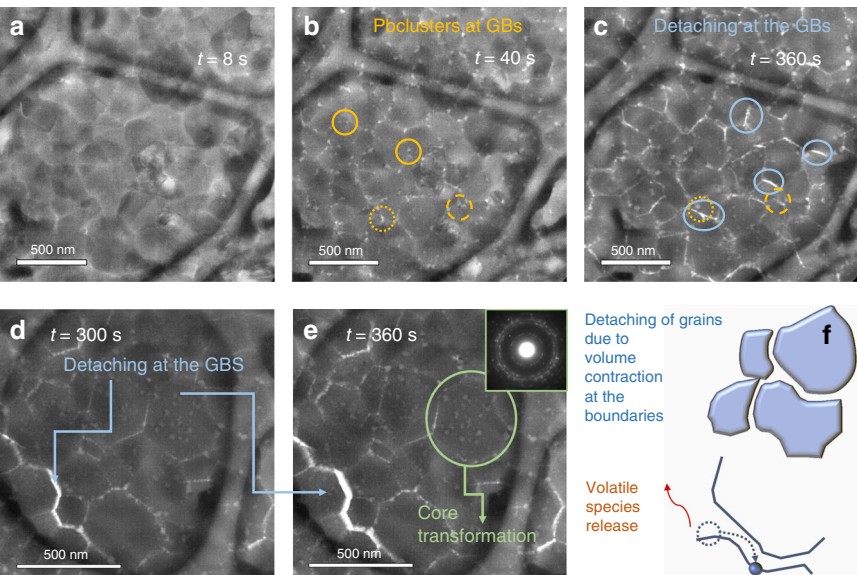

**Fig. 7** Time resolved transformation pathway of a 150 nm-thick sublimated MAPbI$_3$ layer. **a–e** Plan-view TEM images taken along the labelled timescale. The sequence from (**a–c**) is taken at fixed sampled area. Images (**d**, **e**) are taken on another region of the sample and used to highlight the progression of the detachment of a raw of grains. **d** also shows a change in the inner grain morphology (white objects are empty spaces) to 6H-PbI$_2$, as further shown by the EDP in the inset. For each images the acquisition time was 8 s. Electron dose rate was 1 e$^-$/A$^2$ s. **f** Schematic on the detaching phenomenon along a row of grains as a consequence of the GBs shrinkage due to MAPbI$_3$ consumption

Pb-clustering is a surface phenomenon that can occur in evaporated as well as in solution deposited layers under external perturbation. It arises from aggregation of Pb-related defects left at the boundary of MAPbI$_3$ grains after volatile species release as occurs in any kind of dissociation process. Perturbation of equilibrium can be triggered by electron irradiation, even at low dose, in conjunction with morphological inhomogeneities (e.g., grain curvature, triple GBs, and thinned boundaries) and/or local stoichiometric inhomogeneities. High-grain curvature and/or small grain size, promoting Pb atomic diffusion and aggregation, can even lead to oversize clustering, in thin layers. The description of progressive MAPbI$_3$ consumption, preferentially at the GBs, by Pb aggregation also frames the detaching phenomenon observed in our work and in the literature[42].

The effect of the Pb-clustering on the grain surfaces is dramatic in progressively feeding on the hosting matrix. It can occur with similar consequences under any other kind of external perturbation that creates volatile species and defects or an atomic gradient, i.e., under action of environmental species, electric field, light and temperature. Pb-clusters migration without any applied electric field addresses a catalytic action of the clusters in collecting lead atoms from the disordered MAPbI$_3$ boundaries. Pb aggregation could even occur in a complete device architecture as boundaries are created with the ETL and HTL layers. Ionic migration under operation, unbalancing the local atomic equilibrium, could similarly and indirectly promote Pb-aggregation.

Pb-clustering is limited by the concomitant but slower transformation of the core of the MAPbI$_3$ grains to PbI$_2$. This implies that Pb/PbI$_2$ is more stable than Pb/MAPbI$_3$ interfaces. Phase transition of the MAPbI$_3$ grain core to PbI$_2$ is a time-dependent process that deserves further focused investigation. It was here explored by in situ (TEM) and ex situ experiments (XRD). MAPbI$_3$ core progressively converts into 6H-PbI$_2$ in the TEM-transformed sample; 2H-PbI$_2$ is instead generated under air conditions. 6H-PbI$_2$ has a higher structural affinity to MAPbI$_3$ and this allows transforming one structure to the other by small atomic adjustments (plus MA–I release), similarly to what occurs in heteroepitaxial couplings. The process results in a fragmentation of each original

perovskite grain into an array of PbI$_2$ nano-domains sharing a common zone axis (i.e., same planes parallel to the grain surface). We can, indeed, argue that this transition is kinetically favoured. The transition to 2H is instead costlier and thus needs to be activated by catalytic species such as water molecules. In the statistics, nevertheless, 6H inclusions into a 2H-dominant matrix can be even found after degradation under air conditions. In the light of those findings, a convenience in piloting a partial transition of MAPbI$_3$ towards the 6H polymorphism is drawn based on their allowed lattice coupling that addresses the use of a thin conformal coverage of 6H-PbI$_2$ to protect the surface of each perovskite grain. On the opposite, degradation to 2H-PbI$_2$ produces sliced domains not structurally connected to the MAPbI$_3$ mother grain, thus excluding the possibility of a conformal coverage.

Controlled surface passivation by 6H-PbI$_2$ could be explored to preserve the grain core and/or mitigate dynamic disorder at the grain surfaces. This would additionally limit Pb-defects clustering at the periphery of the grains.

Finally, the extended timescale for degradation by scaling-up the layer thickness of the MAPbI$_3$ layer reinforces the role of the surface to volume ratio and suggests the total electron dose (e$^-$/cm$^2$) normalised to the grain thickness as a more comprehensive parameter to merge data from different laboratories in a unique benchmark curve for degradation.

## Methods

**Perovskite layer growth**. MAPbI$_3$ layers were grown on a Cu carbon-covered grid by sequential deposition of PbI$_2$ and MAI via physical sublimation from powders[43] at a base pressure of ~2 × 10$^{-2}$ mbar with the crucibles taken at 350 °C and 150 °C, respectively. The procedure allowed avoiding solvent use and skipping any sample preparation for TEM analyses that would modify the structure of the material, with special regards to the preservation of the organic components. A first set of samples was 40-nm thick. A second batch of 150-nm thick layers was grown on Cu-grids simply by prolonging the deposition time and used to frame the role of the surfaces in samples that can be applied to devices. Although we were forced to double the acquisition time during TEM analyses and indeed the total e-dose per frame to allow reliable imaging, the overall description of the sample modification was consistent to what observed in the thin layers such to draw a unique paradigm for the material evolution during time. In both layers (40 and 150 nm thick) no evidence of average stoichiometric discrepancy from the MAPbI$_3$ composition is

found by diffraction and chemical analyses. Large area samples were similarly grown on glass substrates (20 samples) to allow statistical evaluations on the phase transition under air conditions by XRD analyses

**Transmission electron microscopy**. TEM analyses were done in plan-view using a JEOL JEM 2010F microscope operating at 200 kV. The illumination parameters were chosen in order to reduce the dose rate to ~1 e$^-$/Å$^2$ s. Exposure time per frame acquired were 4 and 8 s for the 40- and 150-nm thick layers, respectively. In most cases, frames were sequentially acquired (as the ones in Fig. 1) for time-resolved experiments. In specific experiments, the beam was, on purpose, inter-rupted between two consecutive images (beam intermittence) such to explore eventual phenomena triggered by the beam exposure further evolving after the beam in switched off. Electron Diffraction pattern (EDPs) series were also acquired with the same protocol.

Scanning transmission electron microscopy images were acquired at 200 kV in scanning mode using a high-angle annular dark field detector in Z-contrast configuration

**X-ray diffraction**. XRD patterns were collected by using a D8Discoved (Bruker AXS) diffractometer equipped with a high precision goniometer (0.0001Å), a thin film attachment (long soller slits) and a Cu-K α source (instrumental broadening 0.07°). Probed area = 0.2 cm$^2$.

## Data availability
The data that support the findings of this study are available from the corresponding author upon request.

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

## Acknowledgements
This activity was partially supported by the national project BEYOND NANO Upgrade (CUP G66J17000350007). We want to also acknowledge the bilateral project founded by CNR (Italy) and JSPS (Japan) for supporting the activity on PSC for the years 2018–2019 (CUP B56C18001070005).

## Author contributions
A.A. conceived the experiment and performed the data cross-linked analyses. C.B. car-ried out TEM acquisition and analyses. E.S. prepared the samples and performed the XRD analyses. I.D. and A.L.M contributed to the discussion on polytypes. C.S. supported the project. All authors contributed with fruitful discussion on the experimental results.

## Additional information

**Competing interests:** The authors declare no competing interests.

