## [Peer Review File · Nature Communications]

Reviewers' comments:

Reviewer #1 (Remarks to the Author):

In this manuscript, the authors reported a nanoscale investigation of the degradation products of halide perovskite films. They showed the formation of Pb clusters and their evolution within the grain boundaries of perovskite films.

I found the work highly timing and of impact for the broad community of perovskite solar cells. While there is no direct correlation to the solar cell stability, the fundamental study reported in this manuscript will be highly relevant to provide solutions towards stable perovskite solar cells.

In conclusion, I can recommend the work for publication in the present form.

Antonio Abate

Reviewer #2 (Remarks to the Author):

The same text is found in the attached .docx file for the authors' convenience.

The authors have written an article detailing the effects of vacuum and electron irradiation on the formation of different lead-based compounds in an initial MAPbI₃ thin film. They grew a dense 40 nm film directly on amorphous carbon using physical evaporation at 70 °C. Using very low dose TEM and STEM they observed the formation of pure Pb particles at the grain boundaries, particularly at three-grain intersections. This process was attributed to the vacuum in the TEM, which was confirmed by the appearance of the Pb particles when the sample was left under vacuum for two days without electron beam irradiation.

Under low-dose illumination, the authors observed the formation of the 6H-polytype of PbI₂. This was observed using electron diffraction of a polycrystalline sample, where they noted a shift in the plane spacing, indicating a shift from pure MAPbI₃ to 6H-PbI₂. Furthermore, the PbI₂ was observed to exist as a group of nano-aggregates within the original MAPbI₃ grain boundary, indicating that the overall orientation was conserved. The authors further studied a MAPbI₃ film which had been exposed to humid air, degrading into PbI₂. They described the resulting form of PbI₂ as the 2H-polytype which corresponded well to the electron diffraction pattern they observed. Little to no 6H-polytype was observed in the air-degraded specimen, indicating that air acts as a catalyst to

preferentially form this polytype, which is not energetically favourable to form under the electron beam. The 6H-polytype has a close orientational relationship with tetragonal MAPbI₃, which the authors say explain the preference of this type as a result of electron-beam degradation. They did not observe any further growth of the Pb nanoclusters once the 6H-PbI₂ began forming, and concluded that this was due to a stabilising effect of the 6H-PbI₂.

The authors conclude that the formation of unwanted Pb nanoclusters and perhaps the transition into the unwanted 2H-PbI₂ can possibly be controlled by the controlled passivation of the surface of the thin film with a layer of 6H-PbI₂.

The work shows skilled and careful control of the electron beam and a good ability to obtain results with a limited electron dose, including the absolutely essential use of very low-dose imaging as well as the use of electron diffraction to infer crystallographic information.

Overall, the article explains some interesting phenomena regarding why the PbI₂ observed under an electron beam is different from that observed after air degradation, but there is a range of elaborations and specifications necessary before this article can be considered for publication.

- The article needs a general and thorough overall editing for language and typos. There are many sentences which are ambiguous or difficult to understand. Some have been highlighted below, but the list is not exhaustive.

- Most of the analysis seems like it comes from a single sample, with a single 'twin' sample prepared to compare between TEM and air degradation. Is this the case, or have these results been reproduced in other samples? It can be difficult to control the stoichiometry of a thermally evaporated film precisely, especially in the case of very thin films like the one studied in this article. The Pb clusters causing holes in the film looks similar to nanoparticle seeds growing. It could indicate an impurity in the initial film caused by non-stoichiometric deposition or by contamination from the evaporation chamber. The possibility of some of the phenomena being described in this article being artefacts induced by the sample preparation should be ruled out by studying additional samples prepared independently. An SEM comparison with a 40 nm film on a solar cell-substrate kept in high vacuum would also be highly useful, and make the findings more relevant to solar cell applications if they are reproduced. A solution-processed film on a solar cell-substrate could also be kept in high vacuum and subsequently studied with SEM to see whether the particle formation is present in solution-processed films as well. If not, the particle formation is likely to be due to the evaporation preparation method.

- Due to the thin nature of the sample, it is possible that the formation of Pb particles observed is due to the very close proximity of all of the atoms to the material surfaces, and that the same results might not be present in a thicker and more solar cell-like film. It would be good to see whether a thicker sample shows the same effects. Furthermore, it should be possible to see the same effects in an SEM, which can give morphological information, which the TEM can not. As such, I would also recommend the authors to include information about a thicker film (around 200-300 nm) in the TEM and a thin and a thicker film in the SEM.

Following is a list of text-specific comments in the current text. The line number refers to the PDF version.

39: Neither reference discusses the quasi-liquid nature of hybrid lead iodide perovskites. Perovskites are highly crystalline as the diffraction patterns in this article show. If the authors refer to the loosely bound halide ions, that should be specified, but even this is not liquid behaviour.

61: Rothmann, Cheng et al. published a paper on this in April 2018 <https://onlinelibrary.wiley.com/doi/abs/10.1002/adma.201800629>. It contains essentially the same information as the paper in reference 26, but is based on analysis of a thin film rather than single crystal which can be compared directly to this work, and is probably more relevant for the article at hand. If no direct mention of the results are included in the final version, it should at least be referred to and the precedence compared to reference 26 be acknowledged.

Figure 1: In the article linked to above, and in reference 25, a different change in the film was observed when exposed to low-dose electron beams. They saw a broadening and thinning of the grain boundaries after extended exposure as well as a loss of intra-grain contrast, and did not observe the formation of Pb particles or holes in the film. They used solution processed films of 300-400 nm thickness. Can the formation of Pb particles be a surface effect due to the thin nature of the film, or was this also observed in thicker films? Can it be due to the presence of elemental lead due to non-stoichiometric evaporation conditions? Solar cells are typically made with films of 10-20 times the thickness, so it would be highly relevant to do the experiment with a thicker film as well, and try to reproduce the results.

97, figure text for Figure 1: The change of contrast from the BF-TEM to the DF-STEM images could be more explicit. Even though it is written out, a non-specialist reader might miss it in the current state and confuse the two imaging modes.

Figure 1 (b): Is this the same area as in (a)? Same film?

What is the total dose at each condition?

What probe conditions were used for STEM? Beam current and dwell time?

115: Has this figure been normalised? There is still some iodine signal in the EDX. What are the EDX conditions? TEM or STEM? Is the iodine signal from the probe having an interaction volume larger than the particle size, or do the particles have some iodine in them? What is the quantified ratio between Pb and I in the EDX data? One would intuitively expect a stronger Pb signal in pure Pb particles than in the bulk material, but the Pb peak intensity does not change. The Pb peak around 2.6 eV is lower in the GB region than in the bulk region. What does this suggest? Is this the only evidence of the particles being pure Pb? More quantification of this claim would be useful and it would be good to label the peaks individually. Pb²⁺ is fairly reactive and it seems strange that it would not react with the I- being released to form the stable PbI₂.

117: Has the area in Figure 1 (b) been exposed to electrons before the image was recorded? Is the clustering caused by the vacuum or the e-beam, or both together?

131: It would be good to include a reference for the claim of the nanoparticles not being visible in XRD due to their size.

132: Does the wording 'it has been observed' refer to this work or to a reference?

133: Is this saturation due to the electron beam or does it happen under vacuum alone as well? How is it possible to distinguish electron and vacuum effects if the MAPbI₃ forms Pb clusters after 4 minutes in vacuum?

The holes in Figure 1 (b) seem bigger than those in Figure 1 (a), but the authors state that the Pb particle formation is saturated after four minutes. How does this fit with the larger holes appearing after two days in vacuum? Does the electron beam stop the formation of the particles? It would be useful to keep an irradiated film under vacuum for an extended period of time to verify whether the radiation stops the particle formation.

Figure 2: By convention, spots in an electron diffraction pattern do not have brackets around them. All mentions of spots, and indexing on the diffraction pattern, should, for example, be written as 224 and not (224). (224) denotes the physical plane but 224 denotes the spot.

It would be helpful to label the BF image and the schematic as well, so the figure contains from (a) to (f) instead of from (a) to (d).

Are the DPs taken from the same area?

What is the time between each pattern?

What is the total dose at each pattern?

Which PbI₂ plane is at 0.317 nm?

Does the white and yellow texts in (c) and (d) refer to the spots marked by the arrows?

Negative crystal coordinates are by convention denoted with a bar on top of the number, not a minus in front.

Cheng et al. showed the first example of a pristine DP in ref. 25. This DP changed quickly but visibly. It is difficult to see from the ring DPs alone whether the structure is fully pristine. It would be good to include a pristine single grain DP to verify the state.

149-150: Does 'figure 1' refer to Figure 1 (a) or (b), or both?

Does the interior of the grains fragment into small dots or the surface? BF-TEM is a bulk technique, it cannot distinguish surface properties, only (thin) bulk properties.

154-158: Are the diffraction patterns taken in the area circled by the yellow circle in Figure 2? Were they collected at the same time as the BF images in Figure 1? If so, time $t=0$ is at least 4 seconds and

not 0 seconds. If the diffraction patterns were collected after the initial exposure, $t=0$ is actually 4 minutes. It is difficult to see from the DPs that there is no PbI_2 present. The DP in Figure 2 (a) contains many spots in between the highlighted rings. Could some of them correspond to PbI_2 ?

164-166: This phenomenon was first described by Cheng et al. in the previously mentioned article.

169: Does poly-layer refer to the polycrystalline nature of the film? A poly-layer is a silicon semiconductor device expression.

170-172: Does this mean 'polycrystalline films, which are better suited for solar cell applications than single crystals, need to be treated with extra care because of the tendency for the MAPbI_3 to transform into Pb at the grain boundaries and the surfaces, and the large surface-to-volume ratio makes this more likely'? The sentence is hard to understand.

173-174: What is the evidence for this? That the Pb clusters stop growing once the grains start turning into 6H-PbI_2 ? It is possible that the Pb clusters and the transition of into 6H-PbI_2 happens at the same time, but the clustering happens faster than the transition, depleting its growth conditions rapidly. If there were an excess of lead in the film, the clustering could happen independently of the electron-induced transition into 6H-PbI_2 . It would be an interesting experiment to irradiate the film immediately upon entering the vacuum for an extended period of time to see if the Pb clusters formed while the grain transition took place.

176-178: Does 'plan-view image figure 2' refer to the unlabelled schematic under the blue and yellow rings in Figure 2?

Figure 3: The index at the top of (a) does not match that in Figure 2 (c). Is that on purpose?

Is the DP the same as in Figure 2 (c), or from a different area with the same total dose exposure?

The (a), (b), and (c) labels are ambiguous in their position and could be shifted a bit to the right.

The negative index in (b) should have the bar on top for 1-14.

The position of the (a), (b), and (c) labels are ambiguous in the figure text, it is difficult to tell which part of the text each label corresponds to.

The atomic structure in (c) is rather crowded and difficult to read. It could probably include only a few unit cells and make each unit cell bigger and clearer.

The atoms in (d) are not labelled.

188: Does the text refer to the upper and lower diffraction patterns seen in Figure 3 (b)? If so, it is not specified in the text.

188-191: Does this refer to Figure 3 (b)? Where is the diffraction pattern taken from? What is the electron exposure time and total dose? The spots corresponding to the (112) and (114) planes (and other corresponding planes) seem to be smeared in a slightly polycrystalline way. Is this due to the inclusion of more than one grain, or due to the beam-induced damage? Pristine spots would be expected to be very well-defined.

201, Figure 4 figure text: Negative index should be designated by a bar above.

204: Was this done in another area from that used in Figure 1 and 2? Is the DP the same as in Figure 2 (c), or different?

205+207: Does the word 'slit' refer to the objective aperture in the TEM?

210: 'uniformgrains' needs a space. How long did the full transition take? What was the total dose?

220: Twin can also be a crystallographic term and can be confusing. Perhaps it would be better to simply state that two samples were grown under the same conditions and refer to them as the TEM- and the air-degraded samples.

228: Does the word 'twin' here refer to the TEM-degraded twin?

230: Bar should be above the 3 in the space group.

Figure 5: There are faint spots inside the diffuse ring that seem to be regular. Do they correspond to the 2H-PbI₂? Does the diffuse ring correspond to the amorphous carbon underneath or something polycrystalline? What are the indices of the spots in the DP? What is the zone axis?

Bar should go above the negative index in the figure text.

247: Does 'sliced' mean that the {001} planes are parallel to the substrate? Or perpendicular?

249: Was the large-area XRD done on the air-degraded samples, or the e-beam and vacuum degraded samples as well? If both, how was it ensured that a large enough area was exposed? Is the purpose of this to verify the generally well-understood transition into PbI₂ in air with electron diffraction?

253: Is Figure 6 (a) an example of an XRD pattern? What do the others look like?

Figure 6: The XRD patterns in (a) and (b) are a bit unclear. Which of the patterns were actually measured and which correspond to simulated patterns? Are the black and red lines in the two figures the measured data? If the ambient samples degrade to 2H-PbI₂, why is the prominent (and rectangular?) peak around 25.5 degrees missing in the red line?

Do the patterns correspond only to air-degraded perovskite?

Where are the different temperature conditions shown in the XRD patterns? Is the red and black lines the average of the different temperatures, or are they all identical?

How are the 2H and 6H areas of (c) determined? From the BF image alone? Are the 2H arrows meant to indicate white nano-sized specs of 2H, or do they refer to the entire grain?

Is the TEM image of an air- or electron-degraded film? If electron degraded, what was the total dose? Is it bright field or dark field? The white arrows refer to dark holes, so it is dark field?

The Figure 6 (e) is not referred to in the figure text or the main text.

283-284: 'At the same time, it has been shown that the process is interrupted as soon as PbI_2 is formed', the evidence for this is inconclusive. The two could be happening at the same time based on the results presented here.

290-293: What are the sources for these claims?

303-304: Why does 2H form at all when it is costlier, even with a catalyst? Why is it preferred?

309: Does 'slices' refer to grains?

317: Was the full conversion into MAPbI_3 confirmed with XRD?

How was the final thickness determined?

How was the stoichiometry of the deposition determined and controlled?

What was the deposition rate? The setup in reference 42 has two sensors that measure the rate of deposition of each precursor, which is essential to ensure a stoichiometric deposition. The deposition pressure seems quite high, normally it would be a few orders of magnitude lower. Is this the pressure before deposition, or do the authors use the pressure to monitor the deposition rate?

Was the sample annealed?

322: 'Dark filed' should probably be 'dark field'.

Are the images in Figure 1 DF or BF? There is some confusion as to what is white and what it is dark in Figure 1 (a) and (b). Some clarification of the contrast would be good for the general audience.

STEM involves scanning a focused probe across the surface in a raster setting, and the beam exposure is different from that of TEM and can be difficult to compare. How was the STEM beam current calculated?

Reviewers' comments:

Reviewer #1 (Remarks to the Author):

In this manuscript, the authors reported a nanoscale investigation of the degradation products of halide perovskite films. They showed the formation of Pb clusters and their evolution within the grain boundaries of perovskite films.

I found the work highly timing and of impact for the broad community of perovskite solar cells. While there is no direct correlation to the solar cell stability, the fundamental study reported in this manuscript will be highly relevant to provide solutions towards stable perovskite solar cells.

In conclusion, I can recommend the work for publication in the present form.

Antonio Abate

Dear reviewer, we thank you for the positive evaluation that indeed encourages new fundamental research on the side of the material science. We are still working to gain further insights and to propose operative stabilising solutions.

Reviewer #2 (Remarks to the Author):

We warmly thank the reviewer for the kindness and time he/she spent in the careful reading of the paper. We specially appreciated the expert point of view of the reviewer. We are indeed providing a revised version of the paper and a point-by-point response letter.

We hereafter summarise some main points of the point-by-point response letter.

As a general comment, Pb-clustering is a surface phenomenon. It has been observed in thin layers (40nm) as well as in thicker samples (150nm) deposited by evaporation without clear evidence of average stoichiometric discrepancy from the MAPbI₃ composition. In comparison with the recent literature on TEM-based experiments, some hints on Pb-clustering can be even found in TEM images of solution processed MAPbI₃ published in the first Nat.Comm. paper from Cheng et al. (2017), although its description was beyond the scope of that work. We also found Pb-clustering in samples deposited by solution processing, as reported in our paper ChemPhysChem 2015, 16, 3064 – 3071, although, at that time, the mechanism was not clear enough to be commented in details.

Pb-clustering follows the release of volatile species containing the organic part of the cage and iodine species, as in any kind of dissociation process. Whatever the driving force for Pb-defects formation, we provide a general awareness on the capability Pb atoms to aggregates and move along grain boundaries and surfaces. Once triggered, Pb-clusters can progressively feed the MAPbI₃ matrix. The experiment on a thicker sample (150nm) suggested by the reviewer, allowed demonstrating the pivotal role of the surfaces in the Pb-aggregation process. Aggregation of Pb is attributed to a local equilibrium perturbation that can be even triggered by low electrons dose irradiation with enhanced local effect at the morphological inhomogeneities (e.g. triple grain boundaries, extended defects) and/or in presence of local stoichiometric inhomogeneities, all assisted by volatile species release. Locally, high grain surface curvature and/or small grain size additionally promote Pb atomic diffusion and aggregation that can lead to oversize clustering. According to this, oversize Pb-clustering, occurring in thin layers or in thin regions of thick samples, is not likely to be found in thick layers. The description of progressive MAPbI₃ consumption by Pb aggregation at the grain boundaries also frames a grain detaching phenomenon in agreement with what observed by Cheng and co-workers (2018).

A second main point is the role of the integral dose of the electrons interacting with the sample during the experiment. Cheng and co-workers, in their last TEM-based work (2018) provided a useful benchmark curve, based on EDX data, on the degradation path under low dose conditions. In our experiment, we kept the electron dose rate at 1e⁻⁷/Å²s for TEM and ED acquisitions along the time-resolved phase transition of the MAPbI₃ layer. Following the reviewer suggestion, we labelled the acquisition time for each frame in the revised paper. The reviewer comments, in this respect, gave us the opportunity to reason about the relationship between the integral e-dose and the changes with thickness. Our findings indicate, in both explored cases (40 and 150nm) a surface triggered transformation and an expansion of the timescale for degradation by increasing the layer thickness that further reinforce the role of the surface to volume ratio. The timescale changing with thickness would suggest the total electron dose to be normalised to the grain volume as a parameter to merge data from different laboratories in a unique degradation curve for benchmark of future works.

Finally, following many points raised by the reviewer, the role of vacuum is better elucidated and correlated to our results in the revised text. On the basis of the extra-experiments we did and of the results we already published, we argue that vacuum has a pivotal role in unbalancing the equilibrium of surface reactions that are locally triggered by perturbations of different nature. Electrons are included in this ensemble. All the details and related argumentations can be found along the text.

In all, the paper was deeply revised and the figures modified or improved, likely gaining in clarity and readiness.

The authors have written an article detailing the effects of vacuum and electron irradiation on the formation of

different lead-based compounds in an initial MAPbI₃ thin film. They grew a dense 40 nm film directly on amorphous carbon using physical evaporation at 70 °C. Using very low dose TEM and STEM they observed the formation of pure Pb particles at the grain boundaries, particularly at three-grain intersections. This process was attributed to the vacuum in the TEM, which was confirmed by the appearance of the Pb particles when the sample was left under vacuum for two days without electron beam irradiation.

We here need to anticipate that the contribution of vacuum, a main point of the experiment, will be better discussed and explained in the following point-by-point response and in the revised version of the paper.

Under low-dose illumination, the authors observed the formation of the 6H-polytype of PbI₂. This was observed using electron diffraction of a polycrystalline sample, where they noted a shift in the plane spacing, indicating a shift from pure MAPbI₃ to 6H-PbI₂. Furthermore, the PbI₂ was observed to exist as a group of nano-aggregates within the original MAPbI₃ grain boundary, indicating that the overall orientation was conserved. The authors further studied a MAPbI₃ film which had been exposed to humid air, degrading into PbI₂. They described the resulting form of PbI₂ as the 2H-polytype which corresponded well to the electron diffraction pattern they observed. Little to no 6H-polytype was observed in the air-degraded specimen, indicating that air acts as a catalyst to preferentially form this polytype, which is not energetically favourable to form under the electron beam. The 6H-polytype has a close orientational relationship with tetragonal MAPbI₃, which the authors say explain the preference of this type as a result of electron-beam degradation. They did not observe any further growth of the Pb nanoclusters once the 6H-PbI₂ began forming, and concluded that this was due to a stabilising effect of the 6H-PbI₂.

The authors conclude that the formation of unwanted Pb nanoclusters and perhaps the transition into the unwanted 2H-PbI₂ can possibly be controlled by the controlled passivation of the surface of the thin film with a layer of 6H-PbI₂.

The work shows skilled and careful control of the electron beam and a good ability to obtain results with a limited electron dose, including the absolutely essential use of very low-dose imaging as well as the use of electron diffraction to infer crystallographic information.

Overall, the article explains some interesting phenomena regarding why the PbI₂ observed under an electron beam is different from that observed after air degradation, but there is a range of elaborations and specifications necessary before this article can be considered for publication.

We would like to thank the reviewer for having so precisely summarised the mainstays of the paper. As he/she noticed, we initially moved along the clear path recently signed by Cheng and co-workers that firstly shed light on the possibility to investigate MAPbI₃ layers via TEM analyses. It is not a trivial point. With respect to those pioneering works, our main focus is on the very early stages of the beam-matter interaction to gain insights on the role the surfaces (nucleation, aggregation, evolution of defects). We are convinced that with a proper revision of the paper, on the basis of the point-by-point reasoning of the reviewer criticisms, the messages will be more compelling for the readers.

- The article needs a general and thorough overall editing for language and typos. There are many sentences which are ambiguous or difficult to understand. Some have been highlighted below, but the list is not exhaustive.

Many thanks for your effort. Corrections and general revision were done.

- Most of the analysis seems like it comes from a single sample, with a single 'twin' sample prepared to compare between TEM and air degradation. Is this the case, or have these results been reproduced in other samples?

We repeated the analyses over 3-5 samples, deposited in different runs but under the same deposition conditions. Some other samples were preliminarily sacrificed in the first analyses during setting the acquisition conditions and indeed were not considered. For X-ray diffraction tens of samples were analysed.

It can be difficult to control the stoichiometry of a thermally evaporated film precisely, especially in the case of very thin films like the one studied in this article. The Pb clusters causing holes in the film looks similar to nanoparticle seeds growing. It could indicate an impurity in the initial film caused by non-stoichiometric deposition or by contamination from the evaporation chamber.

This is a main concern we have elucidated during the experiments. To disentangle this point, we crossed results from different techniques. Firstly, TEM images taken as first acquisition frames (see figure1a: t=4sec=integration time) are totally free from clusters. SEM images are equally uncontaminated. Hereafter we report one image (10kV) of a 150nm-thick MAPbI₃ layer on FTO/TiO₂ (a substrate useful for application) , for your knowledge. As a further value added, in spite of the of the substrate roughness, the grains grown by evaporation are well in touch along the GBs.

According to the electron diffraction, also X-ray diffraction provided information on the tetragonal lattice arrangement of the MAPbI₃ layer without additional visible contributions of Pb or PbI₂ that could be trace from the reagents or of unbalanced reaction. Diffraction patterns of MAPbI₃ grown on different substrates are reported in the following figure:

Nonetheless, local discrepancy from the expected stoichiometry is plausible at the surfaces with respect to the bulk of the grains as effects of asymmetric bonding or local disorder (*A. Alberti et al. Nitrogen soaking promotes lattice recovery in polycrystalline hybrid perovskites, Advanced Energy Materials, 2019*). As a matter of fact, a dynamic process of generation and annihilation of defects occurs at the MAPbI₃ surfaces that involves a disordered shell at the periphery of the grains (*S. Masi et al. Chemical Science 2018, 9, 3200-3208*). The periphery/shell of the grains are firstly exposed to environmental gas species and contaminants, and for that they are of huge interest. In this framework, the experiment demonstrates a weakness at the (triple) grain boundaries as being preferential sites for Pb-defects formation and aggregation that we relate to the easy aggregation of defects, whatever the sollicitation that has generated them. The pivotal role of the surfaces in contributing to structurally change the MAPbI₃ layer has indeed paved the effort through in-situ investigation of the early stages.

This main point was better stated through the revised text in many parts of the paper.

The possibility of some of the phenomena being described in this article being artefacts induced by the sample preparation should be ruled out by studying additional samples prepared independently. An SEM comparison with a 40 nm film on a solar cell-substrate kept in high vacuum would also be highly useful, and make the findings more relevant to solar cell applications if they are reproduced.

See SEM shown before on FTO/TiO₂ substrates

A solution-processed film on a solar cell-substrate could also be kept in high vacuum and subsequently studied with SEM to see whether the particle formation is present in solution-processed films as well. If not, the particle

formation is likely to be due to the evaporation preparation method.

In this respect, experiments were already done. We provide a general description within the next point.

-Due to the thin nature of the sample, it is possible that the formation of Pb particles observed is due to the very close proximity of all of the atoms to the material surfaces, and that the same results might not be present in a thicker and more solar cell-like film. It would be good to see whether a thicker sample shows the same effects. Furthermore, it should be possible to see the same effects in an SEM, which can give morphological information, which the TEM can not. As such, I would also recommend the authors to include information about a thicker film (around 200-300 nm) in the TEM and a thin and a thicker film in the SEM.

We thank the reviewer for this highly motivating suggestion. We intended to implement the paper with new data on a thicker sample that more closely represent a material that can be applied. The reason why we started with thin samples is immediately clear to expert in the field, as the reviewer is, since thinner samples are more suited for TEM analyses. Above all, due to the protocol that is specifically applied to limit beam+vacuum effects on the MAPbI₃ structure, the acquisition time must be kept as low as possible to reduce the integral dose. This is nicely explained in ref Rothmann, Cheng et al. 2018 that the reviewer suggested to read and that was gladly cited in the revised text. Nonetheless, experiment on thicker sample have been running, so that we are now able to produce extra-contents for paper improvement.

We investigated a 150nm-thick-layer of MAPbI₃. It was deposited with the same sublimation procedure simply prolonging the deposition time. As done for the 40nm-thick sample, the layer was firstly investigated by SEM and XRD. Similarly to what found in the thin layer, SEM images of the thick layer do not provide any evidence of the early presence or subsequent formation of Pb-clusters. XRD do not show additional contributions except for the specific features of the tetragonal MAPbI₃ lattice. The findings, on one side, suggest the similarity of the two samples; but on the other hand, they plant doubts on the information accessibility at low level/small size of clustering by SEM. Small Pb-clustering at the grain boundary is not easy to be revealed by SEM analyses. TEM, instead bearing integrated information, is able to unveil small clusters even stuck at the deep boundaries of the grains.

The effect of prolonged SEM analyses is a slight mutual detaching of adjacent grains as shown in the following figure for a 150nm-thick sublimated MAPbI₃ layer. The same phenomenon was found by TEM and shown in the revised text.

figure SEM taken on a 150nm-thick sublimated sample of MAPbI₃

A similar effect on MAPbI₃ layers grown by solution processing is found by TEM investigations in the reference paper suggested by the reviewer (Rothmann, Cheng et al. 2018). In full agreement, our TEM data on the 150nm-thick MAPbI₃ layer deposited by sublimation show grain detaching accompanied by the formation of Pb-clustering at the grain boundaries. A new figure is indeed added, with the related comments, to support a general paradigm on the Pb-clustering being a surface phenomenon. Further comments and figures are provided in the next part of this document.

Following is a list of text-specific comments in the current text. The line number refers to the PDF version.

Our warm thank for the careful reading and evaluating the paper. We changed the paper following all the reviewer suggestions.

39: Neither reference discusses the quasi-liquid nature of hybrid lead iodide perovskites. Perovskites are highly

crystalline as the diffraction patterns in this article show. If the authors refer to the loosely bound halide ions, that should be specified, but even this is not liquid behaviour.

It is nowadays accepted that Hybrid perovskites, with particular focus on MAPbI₃, behaves as a quasi-liquid (i.e. soft) matter. The degrees of freedom of the organic cations inside the inorganic cage and their bonding to the anions create this crystal-liquid duality that makes the overall lattice dynamically rearranging at the local scale (I. Deretzis et al., Spontaneous bidirectional ordering of CH₃NH₃⁺ in lead iodide perovskites at room temperature: The origins of the tetragonal phase. *Sci. Rep.*, 2016, 6, 24443). As a consequence, the lattice of Hybrid Perovskites is easily polarizable, has a low deformation potential, has dynamic defects that can be self-healed, is inclined to be modified by external sollicitation (Kiyoshi Miyata et al., Lead halide perovskites: Crystal-liquid duality, phonon glass electron crystals, and large polaron formation *Sci. Adv.* 2017;3:e1701469; *D. Adv. Mater.*, 2018, 30, 1706273)

This concept was better clarified in the introduction of the revised text.

61: Rothmann, Cheng et al. published a paper on this in April 2018 <https://onlinelibrary.wiley.com/doi/abs/10.1002/adma.201800629>. It contains essentially the same information as the paper in reference 26, but is based on analysis of a thin film rather than single crystal which can be compared directly to this work, and is probably more relevant for the article at hand. If no direct mention of the results are included in the final version, it should at least be referred to and the precedence compared to reference 26 be acknowledged.

We thank the reviewer for this suggestion. The paper contains information that are useful to better frame our results.

It is gladly cited in the new version. Besides what already discussed, we notice the analogy of the behaviour at the grain boundaries between the data shown in the reference (figure 2) and our data in the newly added figure 7. In particular, in the new version we describe the progressive grain detaching occurring after prolonged analyses that has been observed in our experiment (sublimated perovskites) as well as in that of Rothmann (solution processed perovskites). The phenomenon was attributed to the progressive shrinkage of the MAPbI₃ grains periphery due to the progressive mass loss occurring at the grain boundaries. As a further information with respect to the Rothmann's paper, we provide evidence that Pb clusters are formed at the GBs as labelled in figure 7, similarly to what occurs in our 40nm-thick layer. The presence of Pb clusters in the recent Rothmann's paper cannot be discriminated from other structural contributions in the images. Nevertheless, traces of Pb-clustering are visible in the TEM images published by Cheng et al. in 2017 (see figure hereafter reported), although their study was beyond the scope of that paper.

figure 7 (has been added in the revised version): see the detaching phenomenon in the lower panels

figure 2 from reference Rothmann, Cheng et al 2018 to show the grain detaching phenomenon (lower panels)

<https://onlinelibrary.wiley.com/doi/abs/10.1002/adma.201800629>.

Figure 1: In the article linked to above, and in reference 25, a different change in the film was observed when exposed to low-dose electron beams. They saw a broadening and thinning of the grain boundaries after extended exposure as well as a loss of intra-grain contrast, and did not observe the formation of Pb particles or holes in the film. They used solution processed films of 300-400 nm thickness.

Can the formation of Pb particles be a surface effect due to the thin nature of the film, or was this also observed in thicker films? Can it be due to the presence of elemental lead due to non-stoichiometric evaporation conditions? Solar cells are typically made with films of 10-20 times the thickness, so it would be highly relevant to do the experiment with a thicker film as well, and try to reproduce the results.

We clearly find the same phenomenon of Pb-clustering in the thick layer (150nm) as described in the new version of the paper. Although far from the scope of the authors, we notice that some traces of Pb clustering at the grain boundaries can be found in ref 25 in solution processed MAPbI₃ layers deposited on Cu-grid for TEM analyses (Nature Communications , 8:14547 (2017); in figure 1 (e.g. d) hereafter reported):

figure by Cheng et al (Nat. Comm. 2017)

This analogy on differently prepared samples leads us to exclude specificity of the deposition process and to draw a general paradigm on the Pb surface clustering under beam+vacuum conditions.

97, figure text for Figure 1: The change of contrast from the BF-TEM to the DF-STEM images could be more explicit. Even though it is written out, a non-specialist reader might miss it in the current state and confuse the two imaging modes.

Figure 1 (b): Is this the same area as in (a)? Same film?
 What is the total dose at each condition?

We thank the referee for this useful hint. Indeed, to avoid misunderstanding or misleading information, we decided to use a BF-TEM image that has the same contrast as the TEM images of figure 1a. Accordingly, also the graphical abstract has been changed with the corresponding picture (same area as in the submitted version but taken in BF-STEM) as hereafter reported:

New graphical abstract:

new figure 1

What probe conditions were used for STEM? Beam current and dwell time?

Following the reviewer's comment and for the sake of simplicity, we decided to change the DF-STEM image with a BF-TEM. Images acquired with different methods could be misleading.

The area is not the same and this explain the oversize of the Pb-clusters. This is not in contradiction with the observation that the grains core transformation to PbI₂ limits the Pb clustering to small size. Instead, large Pb aggregation was observed after beam exposure (at low dose and low magnification) and long-time pumping in vacuum (days). We indeed argue that the electron beam has triggered volatile species release but not grain core transformation to PbI₂. This last transition is slower than the first one and needs a continuous flux of electrons. Without MAPbI₂ grain core transition to PbI₂, the Pb clustering are free to grow by getting a continuous supply from the perovskite grain surfaces.

115: Has this figure been normalised?

The two figures were normalised to the most intense Pb peak

There is still some iodine signal in the EDX. What are the EDX conditions? TEM or STEM? Is the iodine signal from the probe having an interaction volume larger than the particle size, or do the particles have some iodine in them? What is the quantified ratio between Pb and I in the EDX data?

EDX spectra were acquired in scanning mode at the end of the first transition stage without particular care about the beam softness. This is because we were interested just in analysing the composition of the dark aggregates in the images. Quantitative analyses would merit specific effort due to the fast change expected in the stoichiometry, as nicely done by Cheng and co-workers as reported in the following picture. This curve can be used as benchmark for future specific investigations (e.g by EDX) or to locate expected results on the I/Pb ratio on the basis of the total number of electrons per Å² used in the experiments. Some other comments on the role of the thickness in the total dose needed for full transition to PbI₂ will be given in the next answers and in the revised text.

To explain the extra iodine contribution in figure 1a, we have better specified in the revised version that, although the beam size is comparable with the Pb cluster size, contributions from the surrounding matrix cannot be fully avoided. This indeed produces a small iodine peak in the profile.

One would intuitively expect a stronger Pb signal in pure Pb particles than in the bulk material, but the Pb peak intensity does not change. The Pb peak around 2.6 eV is lower in the GB region than in the bulk region. What does this suggest? Is this the only evidence of the particles being pure Pb?

We warmly thank the reviewer since he/she noticed an anomalous contribution at 2.6eV in the EDX spectrum of the bulk region. On the basis of this input, we realised that this extra-contribution is due to Cl contaminations (likely in NaCl form), that can be encountered during analyses at the TEM (chamber or sample handling contaminants). Indeed, we changed the spectrum with a newly acquired in a close region that does not have contaminants. The figure has been accordingly changed.

For the reviewer knowledge, we hereafter show the complete set of data:

Moreover, as other supporting findings on the nature of the dark clusters, Electron-diffraction was done on large Pb-clusters that confirms the structure

More quantification of this claim would be useful and it would be good to label the peaks individually.

Pb²⁺ is fairly reactive and it seems strange that it would not react with the I⁻ being released to form the stable PbI₂.

The reviewer gets a point that is now be better clarified in the revised text. It is indeed expected a high reactivity between Iodine and lead species at equilibrium in bulk conditions. Instead, the phenomenon of iodine release and Pb clustering occurs at the surfaces. In addition, the vacuum acts in changing equilibrium conditions towards the progressive volatilization of species. Lead atoms are indeed more likely to be free to aggregate. The release of volatile species is a phenomenon that we also observed under mild vacuum conditions by in-situ XRD experiments (see Phys. Chem. Chem. Phys., 18, 13413-13422 (2016)). In that case, the focus was on the kinetics of MAPbI₃ degradation to PbI₂ that was faster than under air conditions due to pumping that unbalances the local equilibrium on the surface. The experiment allowed extracting the activation energy for degradation without (and with, by a similar experiment under humid air) catalysts. We better stated this point at pag 9 in the revised version

117: Has the area in Figure 1 (b) been exposed to electrons before the image was recorded? Is the clustering caused by the vacuum or the e-beam, or both together?
see comments in the previous sections

131: It would be good to include a reference for the claim of the nanoparticles not being visible in XRD due to their size.

size effects of uncorrelated small nanoparticles are expected and is independent of the material.

132: Does the wording 'it has been observed' refer to this work or to a reference?

133: Is this saturation due to the electron beam or does it happen under vacuum alone as well?

How is it possible to distinguish electron and vacuum effects if the MAPbI₃ forms Pb clusters after 4 minutes in vacuum? The holes in Figure 1 (b) seem bigger than those in Figure 1 (a), but the authors state that the Pb particle formation is saturated after four minutes. How does this fit with the larger holes appearing after two days in vacuum? Does the electron beam stop the formation of the particles?

see comments in the previous sections

It would be useful to keep an irradiated film under vacuum for an extended period of time to verify whether the radiation stops the particle formation.

On the basis of the required additional experiment we can draw 3 scenarios:

- 1) no Pb-clusters: sample not irradiated and left in vacuum for days
- 2) small Pb-clusters (~10-15 nm) preferentially located at the boundaries of PbI₂ fragmented grains: low dose rate irradiation ($1e^{-}/A^2sec$) and sequential acquisitions (4sec/frame) for minutes and sample left in vacuum
- 3) large Pb-clusters (~30 nm) with the related empty-tail located at the boundaries of MAPbI₃ grains: oversize aggregation of Pb observed in thin layers or in thin regions of thick samples, likely attributed to a local equilibrium perturbation by low electrons dose irradiation (even in low mag) in thin/high curvature or locally stoichiometric inhomogeneous regions of the grain, all under vacuum-assisted/speeded release of volatile species.

We further noticed that in the thicker sample (150nm) oversize aggregation is not likely to occur, and this reinforces the role of the surfaces and, in particular, of the surface to volume ratio in triggering this phenomenon, as pointed out by the reviewer. Small Pb-clustering is instead observed in the 150nm-thick sample similarly to what occurs in the 40nm-thick layer. Ripening of Pb-clustering at the grain boundaries (formed across the whole boundary) is not as easy as expected in the thin layer.

The findings suggest that vacuum alone is not source of instability, but it rather participate to the sample evolution during time. The specific degradation path of the whole sample (to Pb alone or to Pb+PbI₂) depends on the analyses protocol.

Nonetheless, the mainstay of the experiment has unveiled the possible degradation path of the MAPbI₃ to Pb at the surfaces. Local stoichiometric changes or grain shape and curvature (that promote Pb-aggregation) are variables of the preparation procedure and can likely vary from sample to sample and from laboratory to laboratory. Pb aggregation could even occur in the final devices as boundaries are created with ETL and HTL materials. Ionic migration under operation, unbalancing the local atomic equilibrium, could similarly and indirectly promote Pb-aggregation. Once triggered, Pb aggregation is a dynamic process that progressively feeds the MAPbI₃ layer from its surfaces.

Following the reviewer comment, we improved the description in many point of the paper and added new data on thick MAPbI₃ sample.

Figure 2: By convention, spots in an electron diffraction pattern do not have brackets around them. All mentions of spots, and indexing on the diffraction pattern, should, for example, be written as 224 and not (224). (224) denotes the physical plane but 224 denotes the spot.

It would be helpful to label the BF image and the schematic as well, so the figure contains from (a) to (f) instead of from (a) to (d).

Are the DPs taken from the same area?

What is the time between each pattern?

What is the total dose at each pattern?

specified in the caption and in the revised figure

Which PbI₂ plane is at 0.317 nm?

(014), (1-14)

Does the white and yellow texts in (c) and (d) refer to the spots marked by the arrows?

Negative crystal coordinates are by convention denoted with a bar on top of the number, not a minus in front.

we thank the reviewer

Cheng et al. showed the first example of a pristine DP in ref. 25. This DP changed quickly but visibly. It is difficult to see from the ring DPs alone whether the structure is fully pristine. It would be good to include a pristine single grain DP to verify the state.

The overall pristine state is in figure 2a. The sequence of DPs is taken during time as done for imaging. The timescale is reported in the revised version of the paper.

149-150: Does 'figure 1' refer to Figure 1 (a) or (b), or both?

Does the interior of the grains fragment into small dots or the surface? BF-TEM is a bulk technique, it cannot distinguish surface properties, only (thin) bulk properties.

Fragmentation starts occurring from the surface of the grains. In thin layers, this fragmentation proceeds involving the bulk and is completed in around 4 minutes. In thick layers, this takes longer exposure times.

154-158: Are the diffraction patterns taken in the area circled by the yellow circle in Figure 2? Were they collected at the same time as the BF images in Figure 1? If so, time $t=0$ is at least 4 seconds and not 0 seconds. If the diffraction patterns were collected after the initial exposure, $t=0$ is actually 4 minutes. It is difficult to see from the DPs that there is no PbI₂ present. The DP in Figure 2 (a) contains many spots in between the highlighted rings. Could some of them correspond to PbI₂?

Diffraction patterns are acquired with the protocol described in figure 1 (not at the same time) and over an area comparable to the whole image size. About the contribution in the patterns, the figure has been improved for better clarity. We changed the previous sequence with a more detailed one taken in the same area of the sample during time, as labelled in the new version of figure 2 (hereafter reported). The selected sequence depicts the evolution of the sample in its lattice structure, and is indeed used to frame the continuous transformation with the related epitaxial relationship. Accordingly, the figure shows the image representing the final state of this second transformation step (850sec). We believe that the revised figure, that accounts for all the reviewer comments, has gained in clarity and reliability for the reader. Many thanks.

164-166: This phenomenon was first described by Cheng et al. in the previously mentioned article.

169: Does poly-layer refer to the polycrystalline nature of the film? A poly-layer is a silicon semiconductor device expression.

change done

170-172: Does this mean 'polycrystalline films, which are better suited for solar cell applications than single crystals, need to be treated with extra care because of the tendency for the MAPbI₃ to transform into Pb at the grain boundaries and the surfaces, and the large surface-to-volume ratio makes this more likely'? The sentence is hard to understand.

change done

173-174: What is the evidence for this? That the Pb clusters stop growing once the grains start turning into 6H-PbI₂? It is possible that the Pb clusters and the transition of into 6H-PbI₂ happens at the same time, but the clustering happens faster than the transition, depleting its growth conditions rapidly. If there were an excess of lead in the film, the clustering could happen independently of the electron-induced transition into 6H-PbI₂. It would be an interesting experiment to irradiate the film immediately upon entering the vacuum for an extended period of time to see if the Pb clusters formed while the grain transition took place.

As pointed out by the reviewer, it is not excluded that PbI₂ starts locally forming during irradiation while Pb-clustering is occurring fast. It is instead clear that Pb-clustering is the fastest process since it occurs by transformation of the surface and atomic diffusion in disordered regions of the matrix. Pb-clusters are indeed observed by TEM before PbI₂ formation is detected. About the instability of the interfaces, please see what already answered.

176-178: Does 'plan-view image figure 2' refer to the unlabelled schematic under the blue and yellow rings in Figure 2?

It was better clarified and figure 2 changed according to all the reviewer comments:

New figure 2

Figure 3: The index at the top of (a) does not match that in Figure 2 (c). Is that on purpose?
 correction done

new figure 3

Is the DP the same as in Figure 2 (c), or from a different area with the same total dose exposure?
 a smaller area was used to highlight the biunivocal correspondence in the lattice coupling (doublets)

The (a), (b), and (c) labels are ambiguous in their position and could be shifted a bit to the right.

done

The negative index in (b) should have the bar on top for 1-14.

done

The position of the (a), (b), and (c) labels are ambiguous in the figure text, it is difficult to tell which part of the text each label corresponds to.

done

The atomic structure in (c) is rather crowded and difficult to read. It could probably include only a few unit cells and make each unit cell bigger and clearer.

a bit clarified in the figure caption

The atoms in (d) are not labelled.

done

188: Does the text refer to the upper and lower diffraction patterns seen in Figure 3 (b)? If so, it is not specified in the text.

188-191: Does this refer to Figure 3 (b)? Where is the diffraction pattern taken from? What is the electron exposure time and total dose? The spots corresponding to the (112) and (114) planes (and other corresponding planes) seem to be smeared in a slightly polycrystalline way. Is this due to the inclusion of more than one grain, or due to the beam-induced damage? Pristine spots would be expected to be very well-defined.

thanks. This is one main point. Spots are smeared due to the nano-fragmentation of the matrix. This induces a broadening of the intensity distribution for size effects. We specified in the text.

201, **Figure 4** figure text: Negative index should be designated by a bar above.

done

204: Was this done in another area from that used in Figure 1 and 2? Is the DP the same as in Figure 2 (c), or different?

different area for statistics. It is well representative.

205+207: Does the word 'slit' refer to the objective aperture in the TEM?

correction done

210: 'uniformgrains' needs a space. How long did the full transition take? What was the total dose?

thanks

220: Twin can also be a crystallographic term and can be confusing. Perhaps it would be better to simply state that two samples were grown under the same conditions and refer to them as the TEM- and the air-degraded samples.

changes done

228: Does the word 'twin' here refer to the TEM-degraded twin?

no

230: Bar should be above the 3 in the space group.

Figure 5: There are faint spots inside the diffuse ring that seem to be regular. Do they correspond to the 2H-PbI₂?

The extra spots are likely part of a neighbourhood grain since the selected area is slightly bigger than one grain. The hexagonal DF(circlet) is highly representative since it is mostly encountered all over the sampled areas

Does the diffuse ring correspond to the amorphous carbon underneath or something polycrystalline? What are the indices of the spots in the DP? What is the zone axis?

done

Bar should go above the negative index in the figure text.

done

247: Does 'sliced' mean that the {001} planes are parallel to the substrate? Or perpendicular?

parallel

249: Was the large-area XRD done on the air-degraded samples, or the e-beam and vacuum degraded samples

as well? If both, how was it ensured that a large enough area was exposed? Is the purpose of this to verify the generally well-understood transition into PbI₂ in air with electron diffraction?
 Figure changed and more properly commented

new figure 5

253: Is **Figure 6 (a)** an example of an XRD pattern? What do the others look like?

We left many MAPbI₃ samples to fully degrade in air: They, at the end, were analysed by XRD. In the statistics, two paths are found: a) a degradation towards 2H PbI₂, the most frequently observed case; 2) a degradation with the final state being a mixture of polymorphisms. The observation allows concluding that, although the most frequent degradation path in air is towards 2H-PbI₂, a path more similar to what observed in the TEM-degraded sample is also possible in the air-degraded counterpart. This is demonstrated in figure 6c representing a kind of sample (degraded in air and subsequently analysed at the TEM) wherein 6H- and 2H- PbI₂ coexists. Grains of the two typologies are easily distinguishable in the plan-view TEM image thanks to a different morphology. The two XRD patterns shown in figure 6 are representative of all the samples analysed. This was better explained in the revised text.

Figure 6: The XRD patterns in (a) and (b) are a bit unclear. Which of the patterns were actually measured and which correspond to simulated patterns? Are the black and red lines in the two figures the measured data? If the ambient samples degrade to 2H-PbI₂, why is the prominent (and rectangular?) peak around 25.5 degrees missing in the red line?
 the figure has been reorganised for clarity

Do the patterns correspond only to air-degraded perovskite?
yes

Where are the different temperature conditions shown in the XRD patterns? Is the red and black lines the average of the different temperatures, or are they all identical?

How are the 2H and 6H areas of (c) determined? From the BF image alone? Are the 2H arrows meant to indicate white nano-sized specs of 2H, or do they refer to the entire grain?

In our first runs of analyses, electron DP were associated to the different grains. On this basis, at the end, we were confident that the kind of fragmented morphology can be reconducted to 6H according to figure 2.

Is the TEM image of an air- or electron-degraded film? If electron degraded, what was the total dose? Is it bright field or dark field? The white arrows refer to dark holes, so it is dark field?

corrections and improvement done

The Figure 6 (e) is not referred to in the figure text or the main text.
done

283-284: 'At the same time, it has been shown that the process is interrupted as soon as PbI₂ is formed', the evidence for this is inconclusive. The two could be happening at the same time based on the results presented here.

it was commented that one process is faster than the o

290-293: What are the sources for these claims?

303-304: Why does 2H form at all when it is costlier, even with a catalyst? Why is it preferred?

you can find the comments in the revised text

309: Does 'slices' refer to grains?

317: Was the full conversion into MAPbI₃ confirmed with XRD?

How was the final thickness determined?

How was the stoichiometry of the deposition determined and controlled?

What was the deposition rate? The setup in reference 42 has two sensors that measure the rate of deposition of each precursor, which is essential to ensure a stoichiometric deposition. The deposition pressure seems quite high, normally it would be a few orders of magnitude lower. Is this the pressure before deposition, or do the authors use the pressure to monitor the deposition rate?

the deposition was done at higher pressure than in the literature in a less expensive setup to be more appealing for cost-reduction. the system is customised and we did not provide details on purpose. The basic principle is sublimation.

Was the sample annealed?

322: 'Dark filed' should probably be 'dark field'.

thanks

Are the images in Figure 1 DF or BF? There is some confusion as to what is white and what it is dark in Figure 1 (a) and (b). Some clarification of the contrast would be good for the general audience.

change done in the figure

STEM involves scanning a focused probe across the surface in a raster setting, and the beam exposure is different from that of TEM and can be difficult to compare. How was the STEM beam current calculated?

new figure 6

REVIEWERS' COMMENTS:

Reviewer #2 (Remarks to the Author):

The authors have done a great job at expanding the study to including additional thicker samples, and have elaborated very well on all of the points raised in the original review. The study is now very clear and presents an important step towards understanding the different degradation pathways, especially the difference between the different PbI₂ polytypes resulting from the different exposures.

With the editing of a few minor typos I therefore strongly recommend that this article is published.

162: Should the 'per' be 'for'?

269: 'DF' probably refers to the diffraction pattern and should be 'DP'?

310: The authors refer to an objective aperture for dark field imaging, but then refer to it as a slit. It might be better to use the word 'aperture' for consistency.

RESPONSE LETTER

WE WARMLY THANK THE REVIEWER FOR HIS/HER POSITIVE JUDGMENT.
THE PAPER WAS IMPORVED FIRSTLY THANK TO THE DETAILED REVIEWER COMMENTS AND SUGGESTIONS.

REVIEWERS' COMMENTS:

Reviewer #2 (Remarks to the Author):

The authors have done a great job at expanding the study to including additional thicker samples, and have elaborated very well on all of the points raised in the original review. The study is now very clear and presents an important step towards understanding the different degradation pathways, especially the difference between the different Pbl₂ polytypes resulting from the different exposures.

With the editing of a few minor typos I therefore strongly recommend that this article is published.

162: Should the 'per' be 'for'?

269: 'DF' probably refers to the diffraction pattern and should be 'DP'?

310: The authors refer to an objective aperture for dark field imaging, but then refer to it as a slit. It might be better to use the word 'aperture' for consistency.

ALL THE CORERCTIONS WERE DONE